# Sequence Image Datasets Construction via Deep Convolution Networks

**Xing Jin** [1,2], **Ping Tang** [1,*] and **Zheng Zhang** [1]

1   Aerospace Information Research Institute, Chinese Academy of Sciences, Beijing 100094, China; jinxing@radi.ac.cn (X.J.); zhangzheng@aircas.ac.cn (Z.Z.)
2   School of Electronic, Electrical and Communication Engineering, University of Chinese Academy of Sciences, Beijing 100049, China
*   Correspondence: tangping@aircas.ac.cn; Tel.: +86-139-1092-9397

**Abstract:** Remote-sensing time-series datasets are significant for global change research and a better understanding of the Earth. However, remote-sensing acquisitions often provide sparse time series due to sensor resolution limitations and environmental factors such as cloud noise for optical data. Image transformation is the method that is often used to deal with this issue. This paper considers the deep convolution networks to learn the complex mapping between sequence images, called adaptive filter generation network (AdaFG), convolution long short-term memory network (CLSTM), and cycle-consistent generative adversarial network (CyGAN) for construction of sequence image datasets. AdaFG network uses a separable 1D convolution kernel instead of 2D kernels to capture the spatial characteristics of input sequence images and then is trained end-to-end using sequence images. CLSTM network can map between different images using the state information of multiple time-series images. CyGAN network can map an image from a source domain to a target domain without additional information. Our experiments, which were performed with unmanned aerial vehicle (UAV) and Landsat-8 datasets, show that the deep convolution networks are effective to produce high-quality time-series image datasets, and the data-driven deep convolution networks can better simulate complex and diverse nonlinear data information.

**Keywords:** sequence image datasets; adaptive filter generation network; convolution long short-term memory network; cycle-consistent generative adversarial network; UAV dataset; Landsat-8 dataset

## 1. Introduction

Remote-sensing time-series datasets are an important product and can be applied to research and applications in global changes, such as vegetation phenology changes, land degradation, etc. The successful application of remote-sensing time-series datasets are significant for earth science to expand the growth to a deeper level and to better understand the Earth [1,2].

Time-series analysis usually requires the data to be dense and has equal time intervals to facilitate the process. However, remote-sensing acquisitions often provide sparse time series due to sensor resolution limitations and environmental factors [3], such as cloud noise for optical data. In this case, it is difficult to conduct time-series analysis and construct remote-sensing sequence image datasets.

A conventional method to solve missing data is a linear fitting method, and only uses 1D data for transformation. Seaquist et al. [4] used the ordinary kriging (OK) method to improve the accuracy of a normalized difference vegetable index (NDVI). Berterretche et al. [5] used a spatial interpolation method to interpolate leaf-area index (LAI) data. Bhattacharjee et al. [6] compared the accuracy of remote-sensing data using different spatial interpolation methods and concluded that the accuracy of OK was better than inverse distance weight (IDW). The above methods can use the spatial information of

remote-sensing images for spatial transformation, and cannot use temporal information of remote-sensing images for temporal transformation.

Zhou et al. [7] used NDVI data of moderate-resolution imaging spectroradiometer (MODIS) satellite to conduct simulation experiments and evaluate the Savitzky–Golay (SG) filtering [8] and harmonic analysis of time-series (HANTS) model [9] refactoring effect at different time intervals. According to the daily harmonic changes of land surface temperature (LST), Crosson et al. [10] used the LST data of MODIS Terra and Aqua to repair missing LST points by harmonic analysis. The above methods provide better 1D data fitting for the construction of time-series data at different time intervals and are not suitable for the transformation of high-dimensional time-series data.

The emergence of the enhanced spatial and temporal adaptive reflectance fusion model (ESTARTFM) [11], spatial and temporal adaptive reflectance fusion model (STARTFM) [12], a virtual image pair-based spatio-temporal fusion (VIPSTF) [13], and global dense feature fusion convolutional network [14] had provided ideas for research on sequence construction. These models can obtain high spatio-temporal resolution fusion data, but they cannot elaborate on the spatio-temporal evolution of sequence images and rely heavily on the situation of the original data itself.

Remote-sensing sequence images are a kind of short-range complex and nonlinear 2D data [15], and spatio-temporal information must be considered during construction. It is difficult to construct 2D images in the same way that construction is applied to 1D data. It is difficult to use the conventional linear fitting method for construction. The deep learning method can better simulate complex and diverse nonlinear data information, so it is a better choice for time-series image transformation.

Yuval et al. [16] presented benthic mapping and accelerated segmentation through photogrammetry and multi-level superpixels, and showed the accuracy of repeated surveys using orthorectification and sparse label augmentation. This approach is appropriate for any person who is interested in using photogrammetry for ecological surveys, especially diver-based underwater surveys. Kalajdjieski et al. [17] proposed a novel approach evaluating four different architectures that utilize camera images to estimate the air pollution in those areas. These images were further enhanced with weather data to boost the classification accuracy. The proposed approach exploits the deep learning method combined with data augmentation techniques to mitigate the class imbalance problem. In principle, this approach is suitable for every application where image data is correlated with sensor data, and using them in combination can be beneficial for the learning task. In particular, the adoption of this method in multi-class classification settings with imbalanced classes is particularly beneficial. Wang et al. [18] proposed an unsupervised data augmentation and effective spectral structure extraction method for all hyperspectral samples. This method not only improves the classification accuracy greatly with the newly added training samples but also further improves the classification accuracy of the classifier by optimizing the augmented testing samples. However, as discussed previously, these data augmentation methods cannot make full use of the spatio-temporal features of the original data, and the transfer learning ability of its mapping model is weak. So these methods are not well-suited for remote-sensing time-series data transformation.

Long et al. [19] used frame interpolation as a supervision signal to learn the Convolutional Neural Network (CNN) models for optical flow. However, their main target is optical flow and the interpolated frames tend to be blurry. Liu et al. [20] developed a CNN model for frame interpolation that had an explicit sub-network for motion estimation. Their method obtains not only good interpolation results but also promising unsupervised flow estimation results on KITTI 2012. Niklaus et al. [21] used a deep fully convolutional neural network to estimate spatially adaptive 2D or separable 1D convolution kernels for each output pixel and convolves input frames with them to render the intermediate frame. The convolution kernel can capture local motion between input frames and the coefficients for pixel synthesis. However, as discussed previously, these CNN-based single-frame interpolation methods are not well-suited for multi-frame interpolation.

Taking into account the limitations of the above conventional methods and video interpolation methods, Pan et al. [22] used the convolutional long short-term memory network to conduct the prediction of NDVI. This method can perform prediction based on the state information of time-series, and it belongs to the method of multi-scene transformation. However, its predicted results are affected by the length and missing degree of sequence images. Zhu et al. [23] adopted the cycle-consistent generative adversarial network to conduct style conversion between different pictures. This method can perform bidirectional transformation between different pictures, and its generated results are not affected by the missing degree of sequence images. However, the spectral transformation ability of this method is lower than other deep convolution networks and its results are affected by the noise degree of sequence images.

To improve the accuracy of remote-sensing time-series data transformation, this paper is inspired by the idea of Niklaus, Pan, and Zhu et al., and employs the advantage of different deep convolutional neural networks to make up for the missing remote-sensing data. The major novelty of this paper can be summarized as follows:

- We are one of the first attempts to use the advantage of the different convolution networks to conduct spectral transformation for filling in missing areas of sequence images and produce full remote-sensing sequence images.
- We combine the temporal and spatial neighborhood of sequence images to consider the construction of scene-based remote-sensing sequence images. It does not rely on other high-temporal-resolution remote-sensing data, and only construct datasets based on the sequence itself. This provides a new idea for the construction of remote-sensing datasets.

This paper shows that the data-driven models can better simulate complex and non-linear spectral transformation, then get the generated result based on this transformation. High-quality remote-sensing sequence datasets can be produced using different deep convolution networks. This enriches the research and development of the remote-sensing field.

The rest of the paper is organized as follows. Section 1 reviews related studies regarding remote-sensing time-series data transformation. Section 2 describes the experimental datasets and different deep convolution networks. Section 3 presents the experiments and results, including a generalization of deep convolution networks in the time dimension, visual comparisons, and quantitative evaluation of constructed datasets with single and multiple networks. Section 4 discusses the influence of hyper-parameters on the transformation result. Section 5 concludes the paper.

## 2. Materials and Methods

### 2.1. Experimental Datasets

This paper uses unmanned aerial vehicle (UAV) datasets and Landsat-8 datasets for experiments. Figure 1 shows the location of the UAV datasets. Figure 2 shows the location of the Landsat-8 datasets.

UAV datasets are located in Sougéal marsh (western France, 48.52°N, 1.53°W), which is part of the long-term socio-ecological research (LTSER) site Zone Atelier Armorique. This site is a large flooded grassland of 174 hectares located in the floodplain of the Couesnon River, upstream of Mont-Saint-Michel Bay [24]. The projection-type is France Lambert-93. The spatial resolution is 0.02 m. The number of bands is 4: green, red, red-edge, and near-infrared.

Landsat-8 datasets are located in the southeast of Gansu Province (34.73°N, 105.50°E), which is 36 rows and 129 paths in the Worldwide Reference System (WRS-2). The images of these datasets have a small amount of cloud noise in some months. The projection-type is Universal Transverse Mercator (UTM). The spatial resolution is 30 m. The number of bands is 7: coastal, blue, green, red, near-infrared, short-wave infrared-1, and short-wave infrared-2.

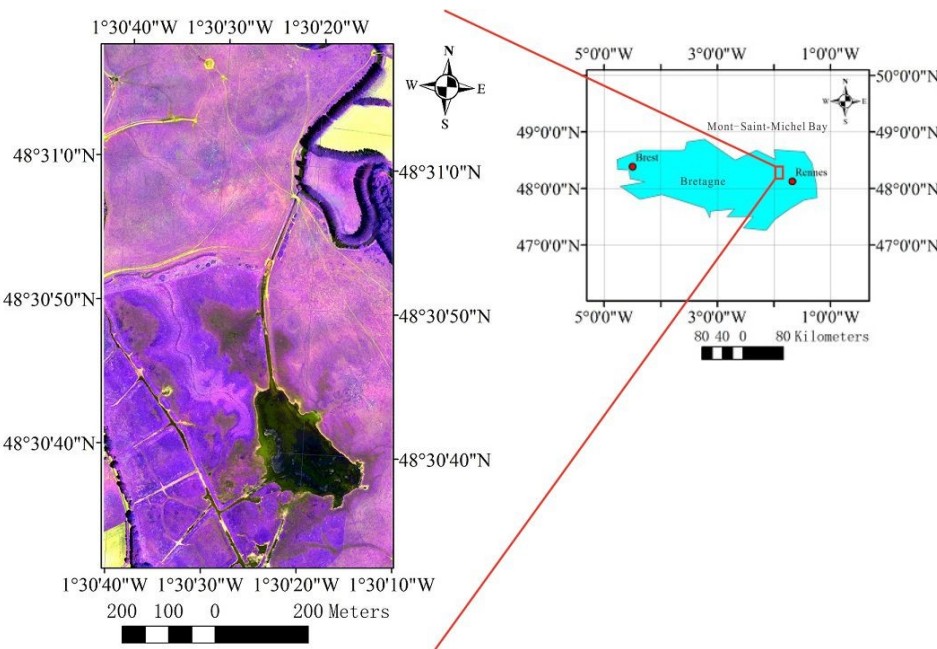

**Figure 1.** Location of unmanned aerial vehicle (UAV) dataset: 1, 2, 3 band composite.

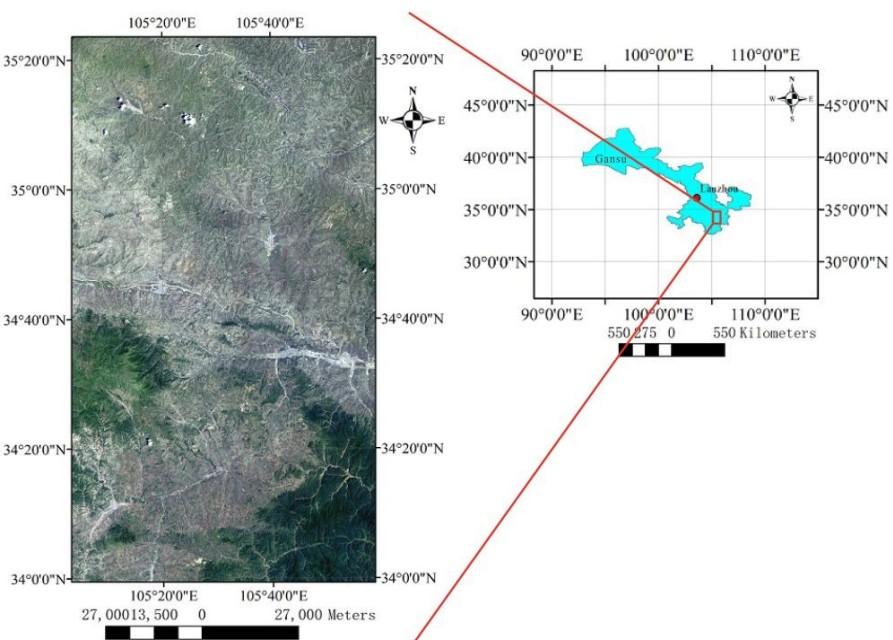

**Figure 2.** Location of Landsat-8 dataset: 5, 4, 3 band composite.

### 2.2. Architecture of Networks

2.2.1. AdaFG Network

Given two images $I_{t_1}$ and $I_{t_2}$ at different times with the same spatial resolution, it is reasonable to assume the image ($I_{estimated}$) at a certain time can be estimated by $I_{t_1}$ and $I_{t_2}$. As is shown in Equation (1):

$$I_{estimated} = b_1(x, y) * K_1(x, y) + b_2(x, y) * K_2(x, y) \qquad (1)$$

In Equation (1), $b_1(x, y)$ and $b_2(x, y)$ are the patches centered at $(x, y)$ in $I_{t_1}$ and $I_{t_2}$, and $K_1(x, y)$ and $K_2(x, y)$ are a pair of 2D convolution kernels; note that $*$ denotes a local convolution operation. The pixel-dependent kernels $K_1$ and $K_2$ capture both motion and re-sampling information required for transformation. The 2D kernels, $K_1$ and $K_2$, could be

approximated by a pair of 1D kernels. That is, $K_1$ could be approximated as $k_{1,v} * k_{1,h}$ and $K_2$ could be approximated as $k_{2,v} * k_{2,h}$. Under this assumption, the main task is to estimate each separable 1D kernel parameter $k_{1,v}, k_{1,h}, k_{2,v}, k_{2,h}$.

The adaptive filter generation network (AdaFG) consists of a contracting part, an expanding part, a separable convolution part, and a backpropagation part [21]. The architecture of the network is shown in Figure 3.

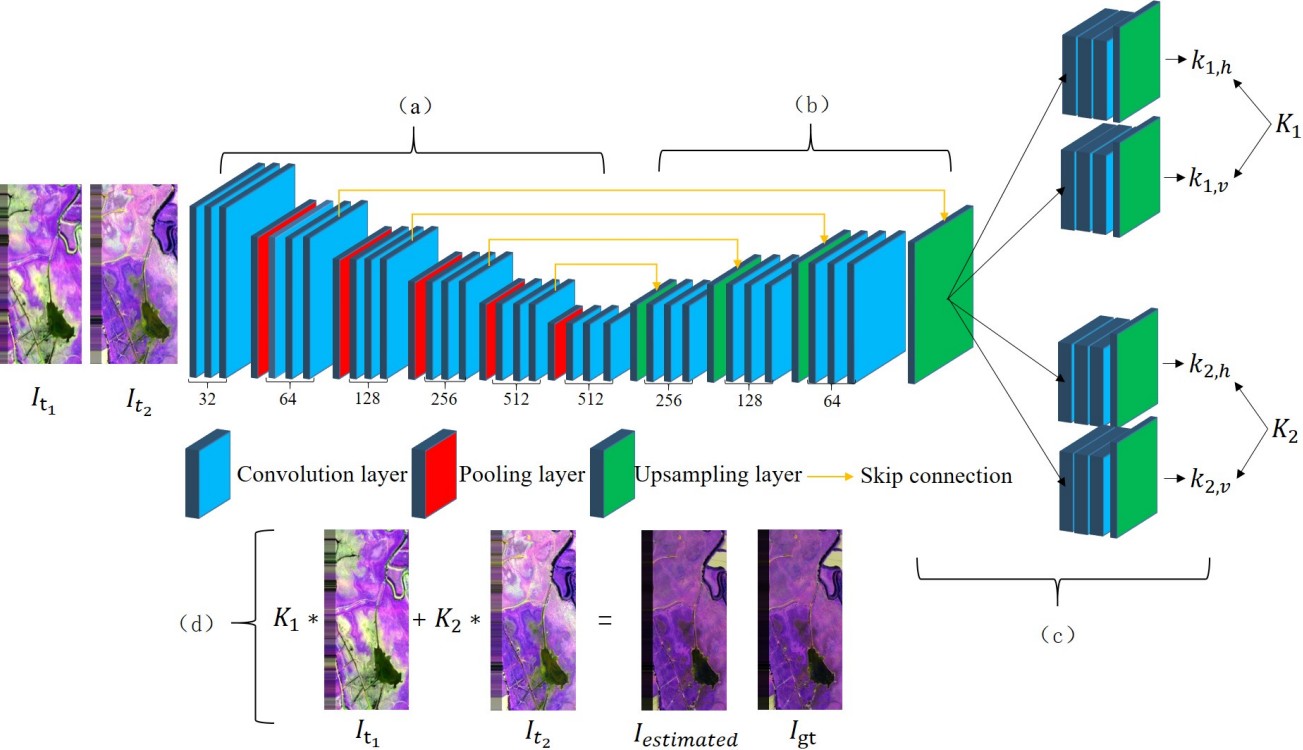

**Figure 3.** Overview of AdaFG architecture: (**a**–**d**) represent feature extracting part, feature expanding part, separable convolution part, and backpropagation part, respectively.

The extracting part (Figure 3a) is used to extract features of training samples. This part mainly contains five convolution layers (*Conv-1, Conv-2, Conv-3, Conv-4* and *Conv-5*) and pooling layers (*Pool-1, Pool-2, Pool-3, Pool-4* and *Pool-5*). The number of filters in each convolution layer is 32, 64, 128, 256, and 512.

The expanding part (Figure 3b) is used to recover the extracted features. This part mainly contains four deconvolution layers (*Deconv-1, Deconv-2, Deconv-3*, and *Deconv-4*) and upsampling layers (*Upsample-1, Upsample-2, Upsample-3*, and *Upsample-4*). The number of filters in each deconvolution layer is 512, 256, 128, and 64.

The separable part (Figure 3c) is used to estimate the separable 1D kernel parameter. This part directs the feature information in the last expansion layer into four sub-networks (*Subnet-1, Subnet-2, Subnet-3*, and *Subnet-4*), with each sub-network evaluating one 1D kernel.

The backpropagation part (Figure 3d) is used to update the weights of the network in each layer. This part updates the weights of the network in each layer according to the error between the generated result ($I_{estimated}$) and the reference image ($I_{gt}$).

The detailed information of each layer of feature maps in the network is shown in Table 1 (input image block W × H = 128 × 128, $n$ represents the number of bands, $k$ represents the size of separable convolution kernel).

**Table 1.** Detailed information of each layer of feature maps in the network.

| Type | Kernel Size | Stride | Padding | Feature Maps (Filters $\times$ **W** $\times$ **H**) |
|---|---|---|---|---|
| *Input* | — | — | — | *2n* $\times$ 128 $\times$ 128 |
| *Conv-1* | 3 $\times$ 3 | 1 $\times$ 1 | | 32 $\times$ 128 $\times$ 128 |
| *Pool-1* | 2 $\times$ 2 | 2 $\times$ 2 | | 32 $\times$ 64 $\times$ 64 |
| *Conv-2* | 3 $\times$ 3 | 1 $\times$ 1 | | 64 $\times$ 64 $\times$ 64 |
| *Pool-2* | 2 $\times$ 2 | 2 $\times$ 2 | | 64 $\times$ 32 $\times$ 32 |
| *Conv-3* | 3 $\times$ 3 | 1 $\times$ 1 | | 128 $\times$ 32 $\times$ 32 |
| *Pool-3* | 2 $\times$ 2 | 2 $\times$ 2 | | 128 $\times$ 16 $\times$ 16 |
| *Conv-4* | 3 $\times$ 3 | 1 $\times$ 1 | | 256 $\times$ 16 $\times$ 16 |
| *Pool-4* | 2 $\times$ 2 | 2 $\times$ 2 | | 256 $\times$ 8 $\times$ 8 |
| *Conv-5* | 3 $\times$ 3 | 1 $\times$ 1 | | 512 $\times$ 8 $\times$ 8 |
| *Pool-5* | 2 $\times$ 2 | 2 $\times$ 2 | | 512 $\times$ 4 $\times$ 4 |
| *Deconv-1* | | | 1 $\times$ 1 | 512 $\times$ 4 $\times$ 4 |
| *Upsample-1* | | | | 512 $\times$ 8 $\times$ 8 |
| *Deconv-2* | | | | 256 $\times$ 8 $\times$ 8 |
| *Upsample-2* | | | | 256 $\times$ 16 $\times$ 16 |
| *Deconv-3* | | | | 128 $\times$ 16 $\times$ 16 |
| *Upsample-3* | 3 $\times$ 3 | 1 $\times$ 1 | | 128 $\times$ 32 $\times$ 32 |
| *Deconv-4* | | | | 64 $\times$ 32 $\times$ 32 |
| *Upsample-4* | | | | 64 $\times$ 64 $\times$ 64 |
| *Subnet-1* | | | | *k* $\times$ 128 $\times$ 128 |
| *Subnet-2* | | | | *k* $\times$ 128 $\times$ 128 |
| *Subnet-3* | | | | *k* $\times$ 128 $\times$ 128 |
| *Subnet-4* | | | | *k* $\times$ 128 $\times$ 128 |

The size of the feature maps has different calculation equations in different layers of the network. The calculation Equations (2)–(5) inside the convolutional layers, pooling layers, deconvolution layers, and upsampling layers are as follows:

$$W_{conv} = \left( W_{input} - K + 2 \cdot P \right) / S + 1 \tag{2}$$

$$W_{pool} = \left( W_{input} - K \right) / S + 1 \tag{3}$$

$$W_{deconv} = \left( W_{input} - 1 \right) \cdot S + K - 2 \cdot P \tag{4}$$

$$W_{upsample} = 2 \cdot W_{input} \tag{5}$$

In the above equations, $W_{input}$ represents the input size of samples, $K$, $S$, and $P$ represent the size of kernel, stride, and padding, respectively. $W_{conv}$, $W_{pool}$, $W_{deconv}$, and $W_{upsample}$ represent the output size of feature maps in convolutional layers, pooling layers, deconvolution layers, and upsampling layers, respectively.

Furthermore, the upsampling layers can be executed in various ways, such as nearest-neighbor, bilinear interpolation, and cubic convolution interpolation [25–27]. We use skip connection [28,29] to let upsampling layers incorporate features from the contracting part of the AdaFG network.

### 2.2.2. CLSTM Network

Convolution long short-term memory network (CLSTM) is an improvement based on the long short-term memory network (LSTM) [22]. It mainly focuses on 2D image data and extracts the spatial characteristics of each time point through the convolution operation. Using the principle of LSTM in the time dimension is to learn long-term characteristics, and then conducts long-term series prediction based on the characteristics. This network can grasp the characteristics of space and time. Compared with the LSTM network, the CLSTM network is an end-to-end network model. The transition of input to state and state to state has a convolution structure [30]. The spatial characteristics are extracted through the convolution operation of the image, and then consider the input of new data and the

output of the previous time point through the input gate and the forget gate. Finally, the output at this time point is determined through the output gate. The cell structure is shown in Figure 4.

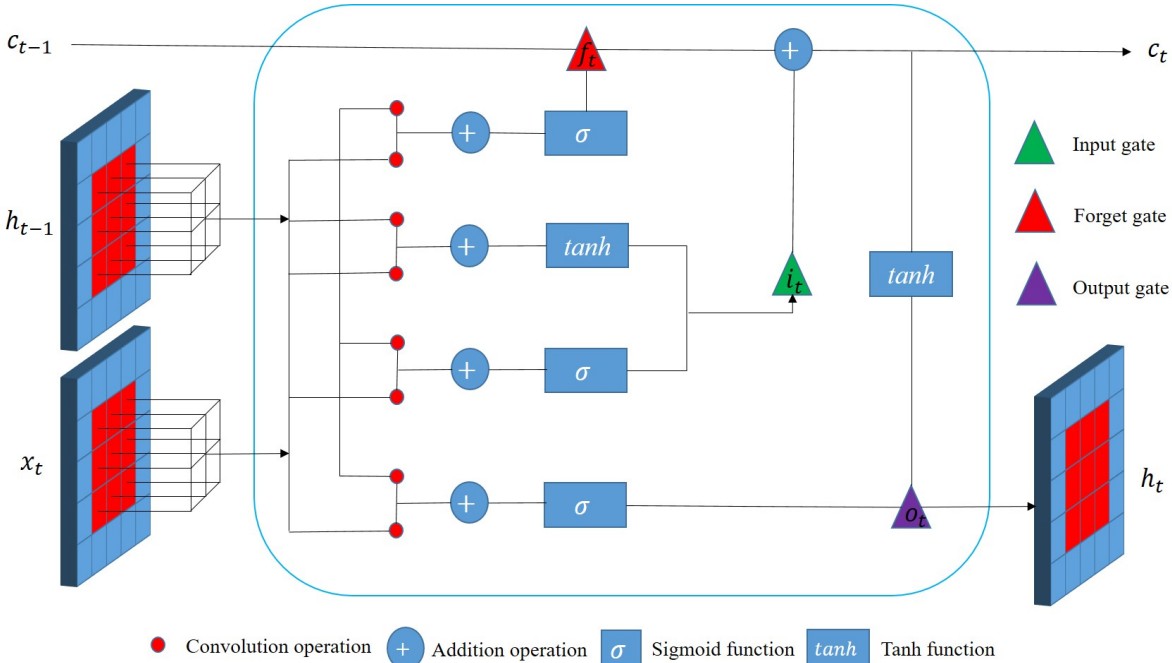

**Figure 4.** Cell structure of CLSTM.

The calculation Equations (6)–(10) inside the CLSTM cell are as follows:

$$f_t = \sigma\left(W_f * [x_t, h_{t-1}] + b_f\right) \tag{6}$$

$$i_t = \sigma(W_i * [x_t, h_{t-1}] + b_i) \tag{7}$$

$$o_t = \sigma(W_o * [x_t, h_{t-1}] + b_o) \tag{8}$$

$$c_t = f_t c_{t-1} + i_t tanh(W_c * [x_t, h_{t-1}] + b_c) \tag{9}$$

$$h_t = o_t tanh(c_t) \tag{10}$$

In the above equations, $*$ denotes convolution operation, $x_t$ represents the input of neurons at $t$, $c_{t-1}$ and $c_t$ represent the cell state of neurons at $t-1$ and $t$, $h_{t-1}$ and $h_t$ represent the output of neurons at $t-1$ and $t$, $W_f$ and $b_f$ represent the weight and bias of the forget gate ($f_t$), $W_i$ and $b_i$ represent the weight and bias of the input gate ($i_t$), $W_o$ and $b_o$ represent the weight and bias of the output gate ($o_t$), $W_c$ and $b_c$ represent the weight and bias of the cell state, and $\sigma$ represents the sigmoid activation function.

### 2.2.3. CyGAN Network

Compared with traditional neural networks, the Generative Adversarial Network (GAN) is an unsupervised learning network. GAN has a collapse problem when conducts style conversion between different pictures. Its model is too uncontrollable for larger pictures and more pixels [31,32]. The cycle-consistent generative adversarial network.

(CyGAN) is an improvement based on the GAN, the purpose of this network is to realize the mutual transformation between a source domain (X) and the target domain (Y) [23]. The structure diagram is shown in Figure 5. CyGAN network mainly contains two mapping functions ($G_{X\to Y}$, $F_{Y\to X}$) and two corresponding adversarial discriminators ($D_Y$, $D_X$). Meanwhile, there are two cycle-consistent losses: forward and backward cycle-consistent losses.

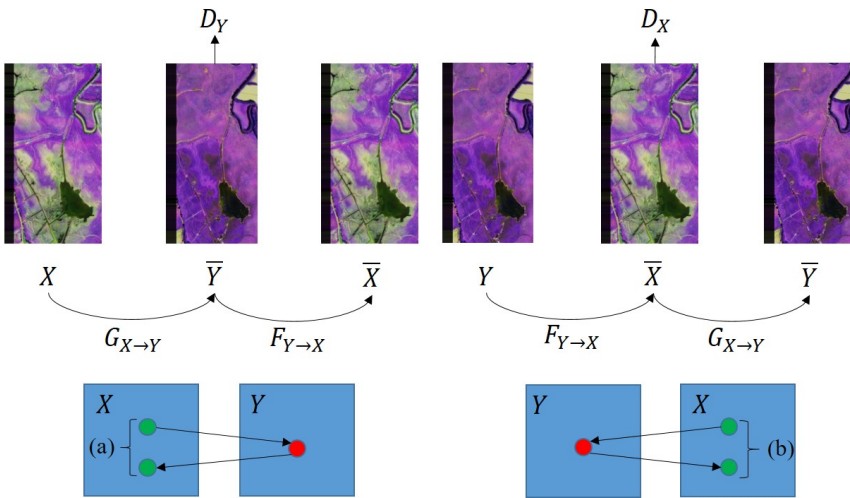

**Figure 5.** Structure diagram of CyGAN: (**a**,**b**) show forward and backward cycle-consistent loss.

According to the CyGAN diagram, CyGAN consists of three parts (forward confrontation target, backward confrontation target, and cycle-consistency loss). The calculation Equations (11)–(14) of the forward confrontation target, backward confrontation target, cycle-consistent loss, and objective function are shown as follows:

$$L_{GAN}(G, D_Y, X, Y) = E_{Y \sim P_{data}(Y)}[log D_Y(Y)] + E_{X \sim P_{data}(X)}[log(1 - D_Y(G(X)))] \quad (11)$$

$$L_{GAN}(F, D_X, Y, X) = E_{X \sim P_{data}(X)}[log D_X(X)] + E_{Y \sim P_{data}(Y)}[log(1 - D_X(F(Y)))] \quad (12)$$

$$L_{cyc}(G, F) = E_{Y \sim P_{data}(Y)}[\|G(F(Y)) - Y\|_1] + E_{X \sim P_{data}(X)}[\|F(G(X)) - X\|_1] \quad (13)$$

$$G*, F* = \arg \min_{G,F} \max_{D_Y, D_X} \{\mu_1 L_{cyc}(G, F) + \mu_2(L_{GAN}(G, D_Y, X, Y) + L_{GAN}(F, D_X, Y, X))\} \quad (14)$$

In the above equations, *X* and *Y* represent the real image in the source domain and target domain, *G(X)* and *F(Y)* represent generated images, $P_{data}(X)$ and $P_{data}(Y)$ represent the distribution of real image in the target domain, ~ represents the obey relationship, *E* represents the mathematical expectation function, $\mu_1$ and $\mu_2$ represent proportional hyper-parameter of the cycle-consistent loss and the confrontation target.

### 2.3. Evaluation Indicator

Our research uses root mean square error (RMSE) and entropy function to evaluate the quality of the generated result: RMSE measures the pixel error between a generated image $I_{estimated}$ and corresponding reference image $I_{gt}$, as defined in Equation (15):

$$RMSE = \sqrt{\frac{1}{W \times H} \Sigma (\|I_{estimated} - I_{gt}\|_2)} \quad (15)$$

The entropy function based on statistical features is an important indicator to measure the richness of image information. The information amount of a generated image $I_{estimated}$ is measured by the information entropy $D(I_{estimated})$, as defined in Equation (16):

$$D(I_{estimated}) = -\sum_{i=0}^{L} P_i ln(P_i) \quad (16)$$

In Equation (16), $P_i$ is the probability of a pixel with a gray value of *i* in image, and *L* is the total number of gray levels (*L* = 255). According to Shannon's information theory [33], there is the most information when there is maximum entropy.

## 3. Results and Analysis

### 3.1. Experimental Strategy

The size of scenes in sequence in both datasets was $3072 \times 5632$. To get enough training samples, all scenes in a sequence were cropped as a block with a size of $128 \times 128$.

This paper mainly conducted three aspects. The first aspect was a generalized application in the time dimension using different deep convolution networks. The second and third aspects were mainly to construct datasets in different time series using single and multiple networks, and this was implemented among multiple sequences. Corresponding experimental data are described in the following paragraphs in detail.

Before describing the experimental data in every aspect, we will first introduce some symbols used below. We represent image scene in different sequences, and $I_{t_i}$, $I'_{t_i}$, $I''_{t_i}$ represent different sequence images acquired at time $t_i$ ($t_i = 1, \ldots, 12$). In AdaFG network, the mapping model is expressed as $f_{[I_{t_1}, I_{t_2}, I_{t_3}]}$, where $[I_{t_1}, I_{t_2}, I_{t_3}]$ represents training image triples, $I_{t_1}$ and $I_{t_2}$ represent the training image pairs, $I_{t_3}$ represents the reference image, $t_1$ and $t_2$ represent the month of training image pairs acquired, and $t_3$ represents the month of the reference image acquired. $f_{[I_{t_1}, I_{t_2}, I_{t_3}]}\left(I'_{t_1}, I'_{t_2}\right)$ represents output image with mapping model $f_{[I_{t_1}, I_{t_2}, I_{t_3}]}$ and input scenes $I'_{t_1}$ and $I'_{t_2}$. In CLSTM network, the mapping model is expressed as $f_{[I_{t_1}, \ldots, I_{t_n}, I_{t_{n+1}}]}$, where $[I_{t_1}, \ldots, I_{t_n}, I_{t_{n+1}}]$ represents training image multi-group, $I_{t_1}, \ldots, I_{t_n}$ represent the training images, $I_{t_{n+1}}$ represents the reference image, $t_1, \ldots, t_n$ represent the month of training images acquired, and $t_{n+1}$ represents the month of the reference image acquired. $f_{[I_{t_1}, \ldots, I_{t_n}, I_{t_{n+1}}]}\left(I'_{t_1}, \ldots, I'_{t_n}\right)$ represents output image with mapping model $f_{[I_{t_1}, \ldots, I_{t_n}, I_{t_{n+1}}]}$ and input scenes $I'_{t_1}, \ldots, I'_{t_n}$. In CyGAN network, the output image is expressed as $f_{[I_{t_1}, I_{t_2}]}\left(I'_{t_1}\right)$, where $f_{[I_{t_1}, I_{t_2}]}$ represents a mapping model trained by training image pair $[I_{t_1}, I_{t_2}]$ with reference image $I_{t_2}$. $I'_{t_1}$ in the brackets represents the input used to generate the output image.

Table 2 shows the sequences used in the first aspect and the dates of all scenes acquired in them. Figure 6 shows the visual effect of training and testing images using different deep convolution networks in the first aspect.

**Table 2.** Name and date of experimental datasets in the first aspect.

| Datasets | Image Names | Image Dates |
|---|---|---|
| UAV | $I_4$ | April 2019 |
| | $I_5$ | May 2019 |
| | $I_6$ | June 2019 |
| | $I_7$ | July 2019 |
| | $I_8$ | August 2019 |
| Landsat-8 | $I'_4$ | April 2014 |
| | $I'_7$ | July 2014 |
| | $I'_2$ | February 2014 |
| | $I'_8$ | August 2014 |
| | $I'_{12}$ | December 2014 |

The second and third aspects mainly constructed remote-sensing sequence datasets with multiple sequences. The remote-sensing sequence here mainly reflected two aspects: non-equidistant missing images in the same sequence and non-equidistant missing images of the same scene in a different sequence. It was difficult to find an analysis method to analyze these sequences in a unified and integrated manner. The number of images in a year was relatively small, and the time interval between images was uncertain.

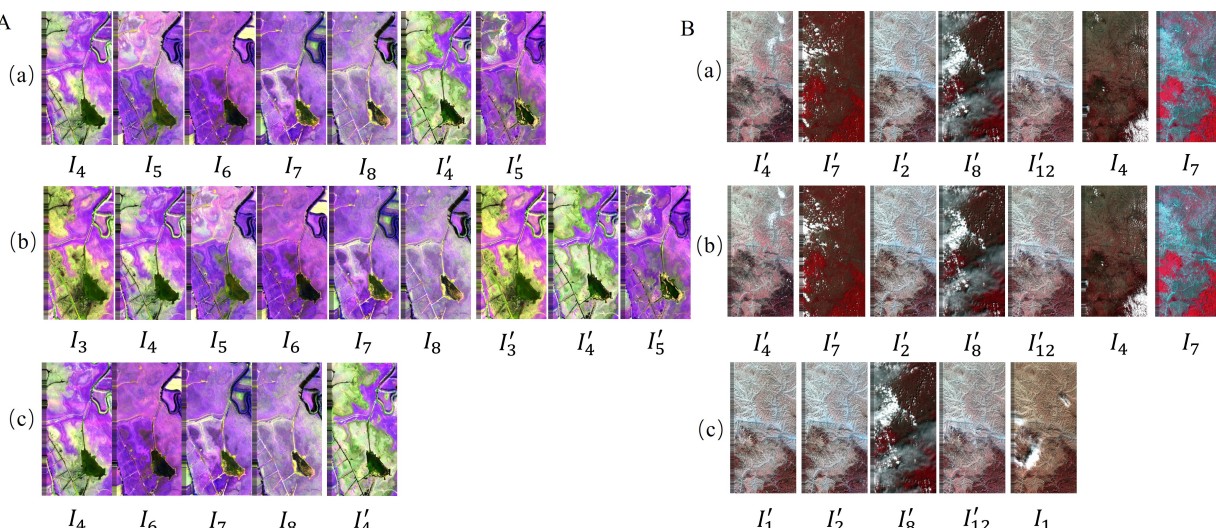

**Figure 6.** Visual effect of training ($I_t$) and testing ($I_t'$) images using different deep convolution networks in the first aspect: (**A**) UAV and (**B**) Landsat-8 images; (**a–c**) show the visual effect using AdaFG, CLSTM, and CyGAN network, respectively.

Figure 7A shows available UAV images from 2017 to 2019 and Landsat-8 images from 2013 to 2015 in the second and third aspects. It is obvious that there are many missing images for the frequency of one image per month. Figure 7B shows one construction strategy with a single network. In UAV datasets, the crimson, red, and light-red points mark the first, second, and third-level generated result, in which training and testing images are listed in Table 3. In Landsat-8 datasets, the dark-green and light-green points mark the first and second generated result, in which training and testing images are listed in Table 4. Figure 7C shows one construction strategy with multiple networks. The red, green, and purple points mark the generated result with AdaFG, CyGAN, and CLSTM network in both datasets, in which training and testing images are listed in Tables 5 and 6.

**Table 3.** Training, testing, and output images with AdaFG in UAV datasets.

| Color. | Training Images | Testing Images | Output Images |
|---|---|---|---|
| Crimson | $I_4, I_5, I_1$ | $I_4', I_5'$ | $f_{[I_4, I_5, I_1]}\left(I_4', I_5'\right)$ |
| | $I_4, I_5, I_3$ | | $f_{[I_4, I_5, I_3]}\left(I_4', I_5'\right)$ |
| | $I_4, I_5, I_6$ | | $f_{[I_4, I_5, I_6]}\left(I_4', I_5'\right)$ |
| | $I_4, I_5, I_7$ | | $f_{[I_4, I_5, I_7]}\left(I_4', I_5'\right)$ |
| | $I_4, I_5, I_8$ | | $f_{[I_4, I_5, I_8]}\left(I_4', I_5'\right)$ |
| | $I_4', I_5', I_9'$ | $I_4, I_5$ | $f_{[I_4', I_5', I_9']}\left(I_4, I_5\right)$ |
| | $I_4', I_5', I_{10}'$ | | $f_{[I_4', I_5', I_{10}']}\left(I_4, I_5\right)$ |
| | $I_4', I_5', I_{12}'$ | | $f_{[I_4', I_5', I_{12}']}\left(I_4, I_5\right)$ |
| Red | $I_7'', I_{10}'', I_{11}''$ | $I_7', I_{10}'$ | $f_{[I_7'', I_{10}'', I_{11}'']}\left(I_7', I_{10}'\right)$ |
| Light-red | $I_4', I_5', I_{11}'$ | $I_4, I_5$ | $f_{[I_4', I_5', I_{11}']}\left(I_4, I_5\right)$ |
| | $I_{10}', I_{11}', I_1'$ | | $f_{[I_{10}', I_{11}', I_1']}\left(I_{10}'', I_{11}''\right)$ |
| | $I_{10}', I_{11}', I_2'$ | | $f_{[I_{10}', I_{11}', I_2']}\left(I_{10}'', I_{11}''\right)$ |
| | $I_{10}', I_{11}', I_3'$ | | $f_{[I_{10}', I_{11}', I_3']}\left(I_{10}'', I_{11}''\right)$ |
| | $I_{10}', I_{11}', I_4'$ | $I_{10}'', I_{11}''$ | $f_{[I_{10}', I_{11}', I_4']}\left(I_{10}'', I_{11}''\right)$ |
| | $I_{10}', I_{11}', I_5'$ | | $f_{[I_{10}', I_{11}', I_5']}\left(I_{10}'', I_{11}''\right)$ |
| | $I_{10}', I_{11}', I_6'$ | | $f_{[I_{10}', I_{11}', I_6']}\left(I_{10}'', I_{11}''\right)$ |
| | $I_{10}', I_{11}', I_8'$ | | $f_{[I_{10}', I_{11}', I_8']}\left(I_{10}'', I_{11}''\right)$ |
| | $I_{10}', I_{11}', I_9'$ | | $f_{[I_{10}', I_{11}', I_9']}\left(I_{10}'', I_{11}''\right)$ |
| | $I_{10}', I_{11}', I_{12}'$ | | $f_{[I_{10}', I_{11}', I_{12}']}\left(I_{10}'', I_{11}''\right)$ |

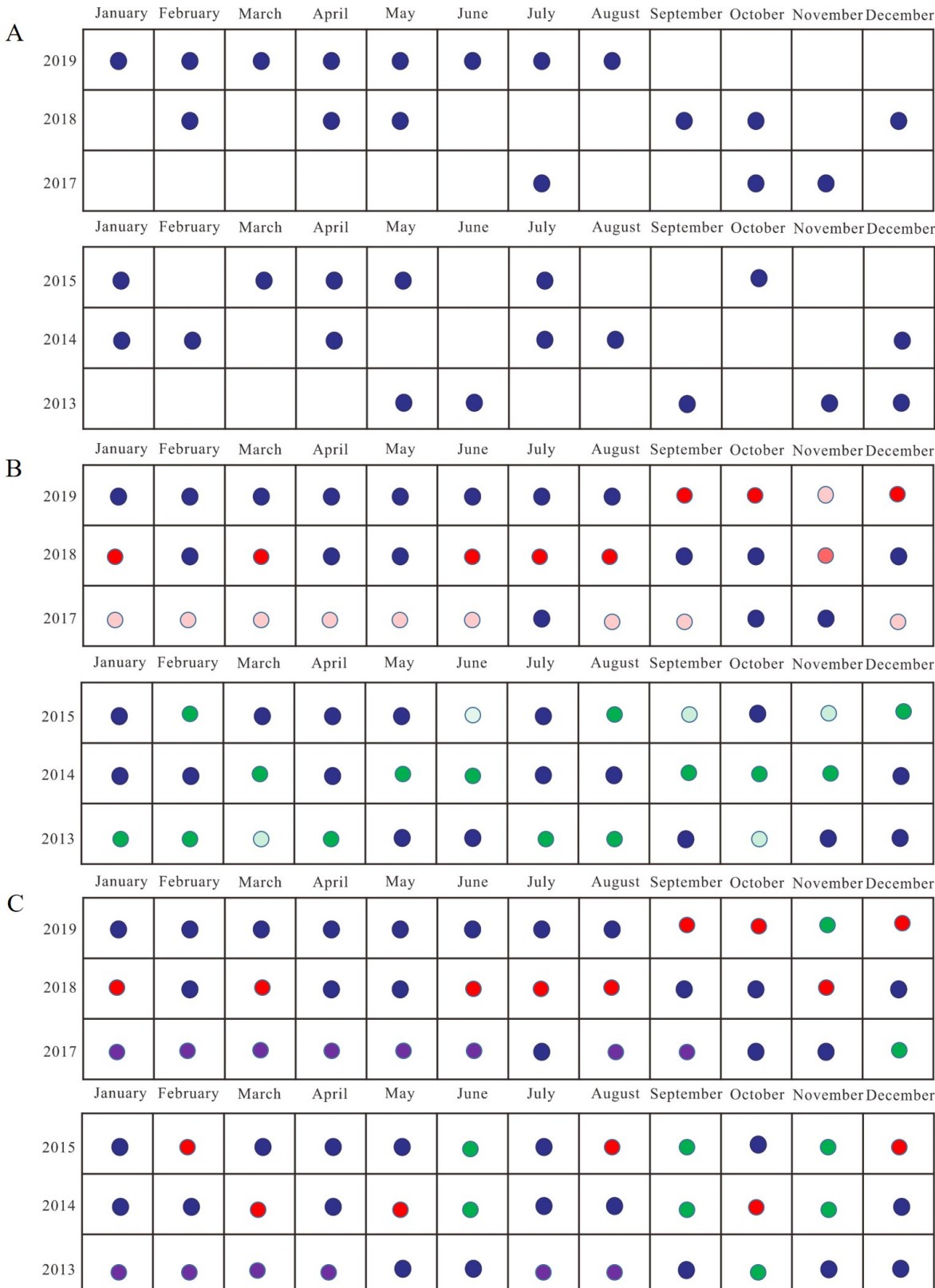

**Figure 7.** Strategy for constructing sequence datasets in the second and third aspect: (**A**) available UAV images from 2017 to 2019 and Landsat-8 images from 2013 to 2015; (**B**) one construction strategy with single network; (**C**) one construction strategy with multiple networks.

**Table 4.** Training, testing, and output images with CyGAN in Landsat-8 datasets.

| Color | Training Images | Testing Images | Output Images |
|---|---|---|---|
| Dark-green | $I'_1, I'_2$ | $I_1$ | $f_{[I'_1, I'_2]}(I_1)$ |
| | $I'_1, I'_8$ | | $f_{[I'_1, I'_8]}(I_1)$ |
| | $I'_1, I'_{12}$ | | $f_{[I'_1, I'_{12}]}(I_1)$ |
| | $I_1, I_3$ | $I'_1$ | $f_{[I_1, I_3]}(I'_1)$ |
| | $I_1, I_5$ | | $f_{[I_1, I_5]}(I'_1)$ |
| | $I_1, I_{10}$ | | $f_{[I_1, I_{10}]}(I'_1)$ |
| | $I''_{12}, I''_6$ | $I'_{12}$ | $f_{[I''_{12}, I''_6]}(I_{12})$ |
| | $I''_{12}, I''_9$ | | $f_{[I''_{12}, I''_9]}(I_{12})$ |
| | $I''_{12}, I''_{11}$ | | $f_{[I''_{12}, I''_{11}]}(I_{12})$ |
| | $I'_{12}, I'_1$ | $I''_{12}$ | $f_{[I'_{12}, I'_1]}(I''_{12})$ |
| | $I'_{12}, I'_2$ | | $f_{[I'_{12}, I'_2]}(I''_{12})$ |
| | $I'_{12}, I'_4$ | | $f_{[I'_{12}, I'_4]}(I''_{12})$ |
| | $I'_{12}, I'_7$ | | $f_{[I'_{12}, I'_7]}(I''_{12})$ |
| | $I'_{12}, I'_8$ | | $f_{[I'_{12}, I'_8]}(I''_{12})$ |
| Light-green | $I'_1, I'_6$ | $I_1$ | $f_{[I'_1, I'_6]}(I_1)$ |
| | $I'_1, I'_9$ | | $f_{[I'_1, I'_9]}(I_1)$ |
| | $I'_1, I'_{11}$ | | $f_{[I'_1, I'_{11}]}(I_1)$ |
| | $I'_{12}, I'_3$ | $I''_{12}$ | $f_{[I'_{12}, I'_3]}(I''_{12})$ |
| | $I'_{12}, I'_{10}$ | | $f_{[I'_{12}, I'_{10}]}(I''_{12})$ |

**Table 5.** Training, testing, and output images with AdaFG, CyGAN, and CLSTM in UAV datasets.

| Networks. | Training Images | Testing Images | Output Images |
|---|---|---|---|
| AdaFG | $I_4, I_5, I_1$ | $I'_4, I'_5$ | $f_{[I_4, I_5, I_1]}(I'_4, I'_5)$ |
| | $I_4, I_5, I_3$ | | $f_{[I_4, I_5, I_3]}(I'_4, I'_5)$ |
| | $I_4, I_5, I_6$ | | $f_{[I_4, I_5, I_6]}(I'_4, I'_5)$ |
| | $I_4, I_5, I_7$ | | $f_{[I_4, I_5, I_7]}(I'_4, I'_5)$ |
| | $I_4, I_5, I_8$ | | $f_{[I_4, I_5, I_8]}(I'_4, I'_5)$ |
| | $I'_4, I'_5, I'_9$ | $I_4, I_5$ | $f_{[I'_4, I'_5, I'_9]}(I_4, I_5)$ |
| | $I'_4, I'_5, I'_{10}$ | | $f_{[I'_4, I'_5, I'_{10}]}(I_4, I_5)$ |
| | $I'_4, I'_5, I'_{12}$ | | $f_{[I'_4, I'_5, I'_{12}]}(I_4, I_5)$ |
| | $I''_7, I''_{10}, I''_{11}$ | $I'_7, I'_{10}$ | $f_{[I''_7, I''_{10}, I''_{11}]}(I'_7, I'_{10})$ |
| CyGAN | $I'_4, I'_{11}$ | $I_4$ | $f_{[I'_4, I'_{11}]}(I_4)$ |
| | $I'_{10}, I'_{12}$ | $I''_{10}$ | $f_{[I'_{10}, I'_{12}]}(I''_{10})$ |
| CLSTM | $I'_{10}, I'_{11}, I'_8, I'_9$ | $I''_{10}, I''_{11}$ | $f_{[I'_{10}, I'_{11}, I'_8]}(I''_{10}, I''_{11})$ |
| | | | $f_{[I'_{10}, I'_{11}, I'_9]}(I''_{10}, I''_{11})$ |
| | $I'_7, I'_8, I'_9, I'_{10}, I'_{11}, I'_{12}$ $I'_1, I'_2, I'_3, I'_4, I'_5, I'_6$ | $I''_7, I''_8, I''_9, I''_{10}, I''_{11}, I''_{12}$ | $f_{[I'_7, \ldots, I'_{12}, I'_1]}(I''_7, \ldots, I''_{12})$ |
| | | | $f_{[I'_7, \ldots, I'_{12}, I'_2]}(I''_7, \ldots, I''_{12})$ |
| | | | $f_{[I'_7, \ldots, I'_{12}, I'_3]}(I''_7, \ldots, I''_{12})$ |
| | | | $f_{[I'_7, \ldots, I'_{12}, I'_4]}(I''_7, \ldots, I''_{12})$ |
| | | | $f_{[I'_7, \ldots, I'_{12}, I'_5]}(I''_7, \ldots, I''_{12})$ |
| | | | $f_{[I'_7, \ldots, I'_{12}, I'_6]}(I''_7, \ldots, I''_{12})$ |

**Table 6.** Training, testing, and output images with AdaFG, CyGAN, and CLSTM in Landsat-8 datasets.

| Networks | Training Images | Testing Images | Output Images |
|---|---|---|---|
| AdaFG | $I'_4, I'_7, I'_2$ <br> $I'_4, I'_7, I'_8$ <br> $I'_4, I'_7, I'_{12}$ | $I_4, I_7$ | $f_{[I'_4,I'_7,I'_2]}(I_4, I_7)$ <br> $f_{[I'_4,I'_7,I'_8]}(I_4, I_7)$ <br> $f_{[I'_4,I'_7,I'_{12}]}(I_4, I_7)$ |
|  | $I_4, I_7, I_3$ <br> $I_4, I_7, I_5$ <br> $I_4, I_7, I_{10}$ | $I'_4, I'_7$ | $f_{[I_4,I_7,I_3]}(I'_4, I'_7)$ <br> $f_{[I_4,I_7,I_5]}(I'_4, I'_7)$ <br> $f_{[I_4,I_7,I_{10}]}(I'_4, I'_7)$ |
| CyGAN | $I''_{12}, I''_6$ <br> $I''_{12}, I''_9$ <br> $I''_{12}, I''_{11}$ | $I'_{12}$ | $f_{[I''_{12},I''_6]}(I'_{12})$ <br> $f_{[I''_{12},I''_9]}(I'_{12})$ <br> $f_{[I''_{12},I''_{11}]}(I'_{12})$ |
|  | $I'_1, I'_6$ <br> $I'_1, I'_9$ <br> $I'_1, I'_{11}$ | $I_1$ | $f_{[I'_1,I'_6]}(I_1)$ <br> $f_{[I'_1,I'_9]}(I_1)$ <br> $f_{[I'_1,I'_{11}]}(I_1)$ |
|  | $I'_{12}, I'_{10}$ | $I''_{12}$ | $f_{[I'_{12},I'_{10}]}(I''_{12})$ |
| CLSTM | $I'_{11}, I'_{12}, I'_7, I'_8$ | $I''_{11}, I''_{12}$ | $f_{[I'_{11},I'_{12},I'_7]}(I''_{11}, I''_{12})$ <br> $f_{[I'_{11},I'_{12},I'_8]}(I''_{11}, I''_{12})$ |
|  | $I'_5, I'_6, I'_7, I'_8\ I'_1, I'_2, I'_3, I'_4$ | $I''_5, I''_6, I''_7, I''_8$ | $f_{[I'_5,\ldots,I'_8,I'_1]}(I''_5, \ldots, I''_8)$ <br> $f_{[I'_5,\ldots,I'_8,I'_2]}(I''_5, \ldots, I''_8)$ <br> $f_{[I'_5,\ldots,I'_8,I'_3]}(I''_5, \ldots, I''_8)$ <br> $f_{[I'_5,\ldots,I'_8,I'_4]}(I''_5, \ldots, I''_8)$ |

### 3.2. Experimental Details

**Hyper-parameters selection:** We used a separate convolution kernel of size 11 in the AdaFG network, used stacks of $3 \times 3$ CLSTM network layer, and set $\mu_1 = 350$, $\mu_2 = 1/32$ in CyGAN network. Meanwhile, we used the mean square error (MSE) as the loss function. The optimizer used in the training was Adamax with $\beta_1 = 0.9$, $\beta_2 = 0.99$, and a learning rate of 0.001. Compared to other network optimizers, Adamax could achieve better convergence of the model [34].

**Time Complexity:** We used the python machine learning library to execute the deep convolution networks. To improve computational efficiency, we organized our layer in computer unified device architecture (CUDA). Our deep convolution networks were able to generate a $256 \times 256$ image block in 4 s. Obtaining the overall scene image (image size $3072 \times 5632$) took about 18 min under the acceleration of the graphics processing unit (GPU) [35].

### 3.3. Experimental Results

#### 3.3.1. Generalization of Networks

Table 7 shows the quantitative evaluation indicator between the generated result and reference image in the first aspect. The table shows that the entropy value produced by using the AdaFG network was higher than that using other networks, and the RMSE [36] value produced by using the AdaFG network was lower than that using other networks in both datasets. Figure 8 shows the visual effect and pixel error between the generated result and reference image using different networks in both datasets and illustrates that the generated result using different networks was close to the reference image. Figure 9 shows the spectral curves between the generated result and reference image using different networks at different coordinates (vegetation, lake, grass, forest) and illustrates that using different networks could maintain better spectral features between the generated result and the reference image in both datasets.

**Table 7.** Quantitative evaluation between the generated result and the reference image.

| Datasets | Generated Results | Reference Images | Entropy | RMSE (Pixel) |
|---|---|---|---|---|
| UAV | $f_{[I_4,I_5,I_6]}\left(I_4', I_5'\right)$ | $I_6$ | 3.658 | 1.128 |
| | $f_{[I_3,I_4,I_5,I_6]}\left(I_3', I_4', I_5'\right)$ | | 3.625 | 1.758 |
| | $f_{[I_4,I_6]}\left(I_4'\right)$ | | 3.592 | 3.585 |
| | $f_{[I_4,I_5,I_7]}\left(I_4', I_5'\right)$ | $I_7$ | 3.355 | 0.952 |
| | $f_{[I_3,I_4,I_5,I_7]}\left(I_3', I_4', I_5'\right)$ | | 3.330 | 1.300 |
| | $f_{[I_4,I_7]}\left(I_4'\right)$ | | 3.305 | 3.077 |
| | $f_{[I_4,I_5,I_8]}\left(I_4', I_5'\right)$ | $I_8$ | 3.482 | 0.954 |
| | $f_{[I_3,I_4,I_5,I_8]}\left(I_3', I_4', I_5'\right)$ | | 3.455 | 1.421 |
| | $f_{[I_4,I_8]}\left(I_4'\right)$ | | 3.426 | 3.690 |
| Landsat-8 | $f_{[I_4',I_7',I_2']}\left(I_4, I_7\right)$ | $I_2'$ | 3.064 | 1.144 |
| | $f_{[I_4',I_7',I_2']}\left(I_4, I_7\right)$ | | 2.975 | 1.537 |
| | $f_{[I_1',I_2']}\left(I_1\right)$ | | 2.905 | 2.694 |
| | $f_{[I_4',I_7',I_8']}\left(I_4, I_7\right)$ | $I_8'$ | 3.830 | 1.170 |
| | $f_{[I_4',I_7',I_8']}\left(I_4, I_7\right)$ | | 3.830 | 1.681 |
| | $f_{[I_1',I_8']}\left(I_1\right)$ | | 3.777 | 5.017 |
| | $f_{[I_4',I_7',I_{12}']}\left(I_4, I_7\right)$ | $I_{12}'$ | 2.994 | 0.929 |
| | $f_{[I_4',I_7',I_{12}']}\left(I_4, I_7\right)$ | | 2.848 | 1.484 |
| | $f_{[I_1',I_{12}']}\left(I_1\right)$ | | 2.796 | 2.629 |

### 3.3.2. Single Network Datasets

In the second aspect, we constructed single network datasets according to the characteristics of different deep convolutions networks and existing images in both datasets. There were no reference images in this aspect. Figure 10 shows the generated results of UAV images using the AdaFG network from 2017 to 2019 and the visual effects of three-level generation according to the construction strategy in Table 3. Figure 11 shows the generated results of Landsat-8 images using the CyGAN network from 2013 to 2015 and the visual effects of two-level generation according to the construction strategy in Table 4. It appears that as the level of generation increased, the spectral features of the generated result became worse. This may be caused by the propagation of pixel error as the level of generation increased.

### 3.3.3. Multiple Network Datasets

In the third aspect, we constructed multiple network datasets to reduce the pixel error of the generated result caused by a single network. There were no reference images in this aspect. Figure 12 shows the generated results of UAV images using AdaFG, CLSTM, and CyGAN network from 2017 to 2019 and the visual effects of the generated results according to the construction strategy in Table 5. Figure 13 shows the generated results of Landsat-8 images using AdaFG, CLSTM, and CyGAN network from 2013 to 2015 and the visual effects of the generated results according to the construction strategy in Table 6. It appears that the visual effects of multiple network datasets are better than a single network.

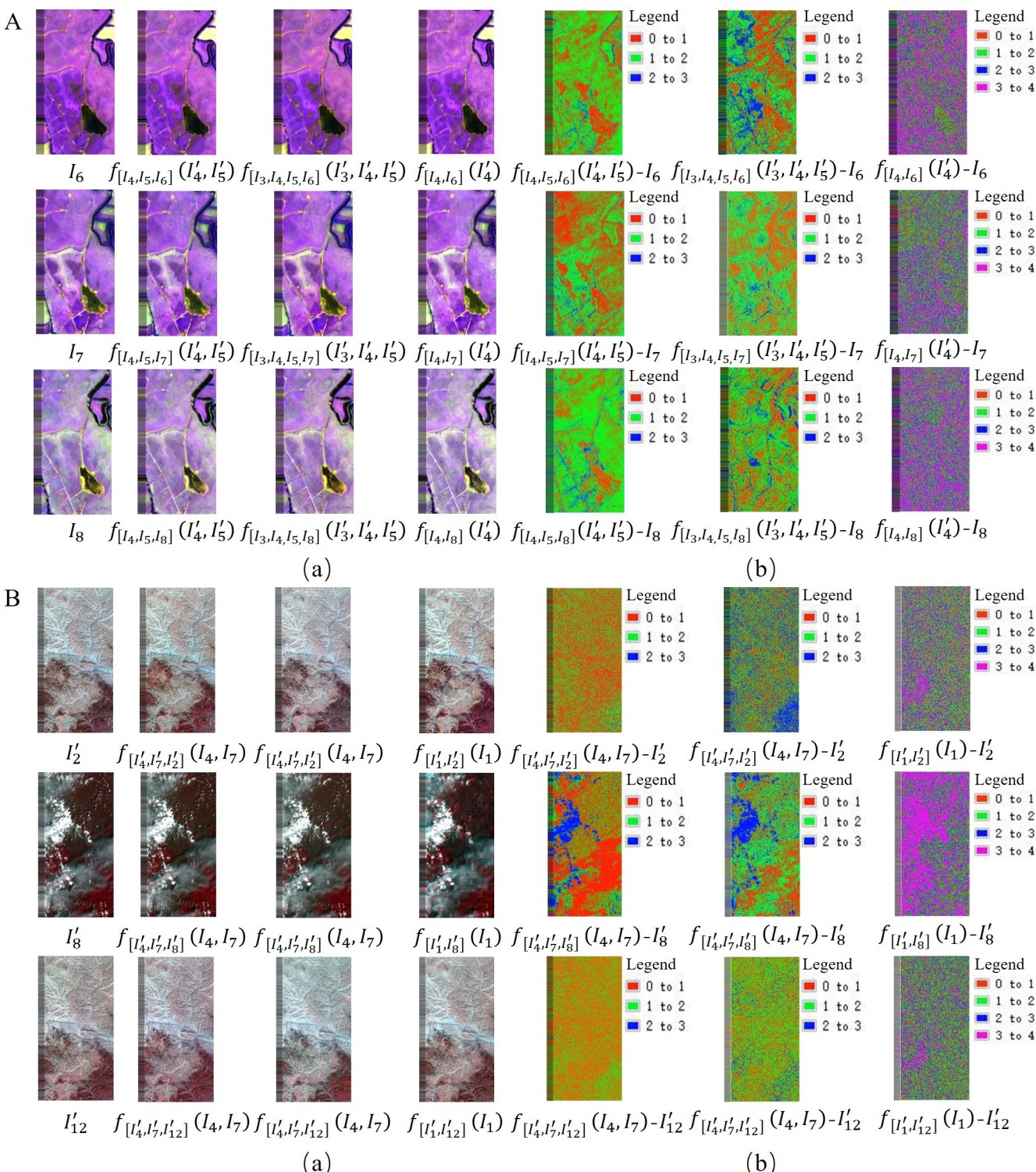

**Figure 8.** (**a**) Visual effect and (**b**) pixel error between generated result and reference image using different deep convolution networks: (**A**) UAV and (**B**) Landsat-8 datasets.

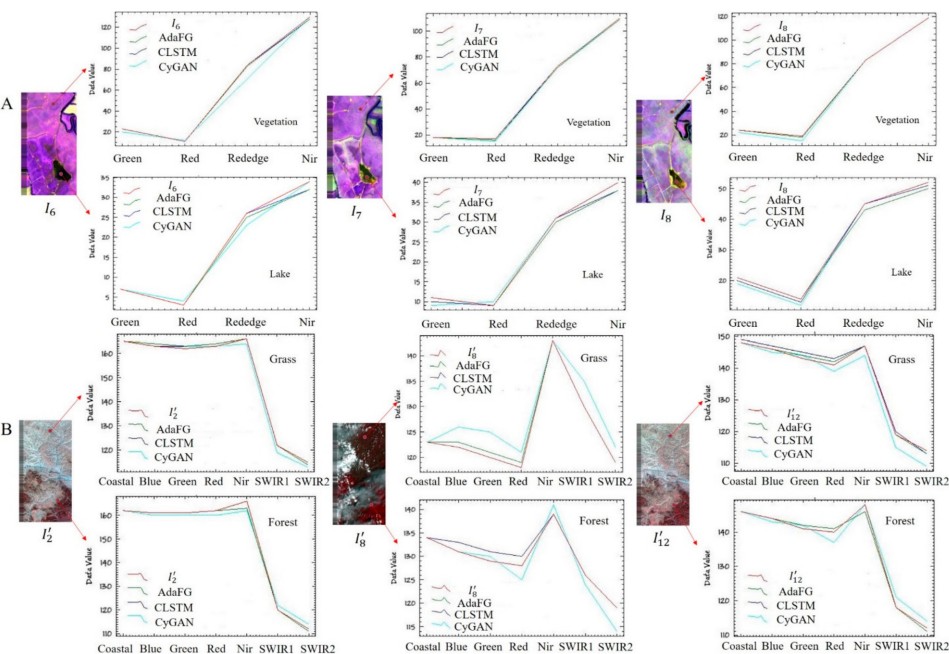

**Figure 9.** Spectral curves between generated result and reference image using different deep convolution networks at different coordinates: (**A**) UAV and (**B**) Landsat-8 datasets.

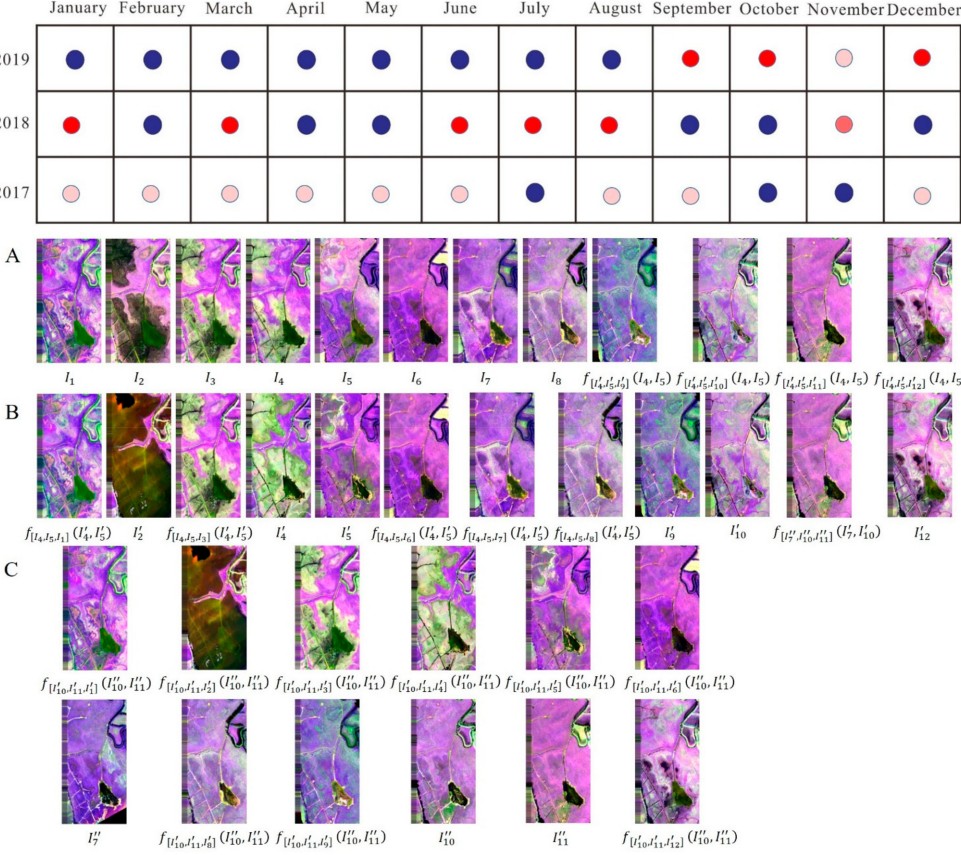

**Figure 10.** Generated results of UAV images using AdaFG network from 2017 to 2019 according to construction strategy in Table 3: existing images and generated results in (**A**) 2019 sequence, (**B**) 2018 sequence, and (**C**) 2017 sequence.

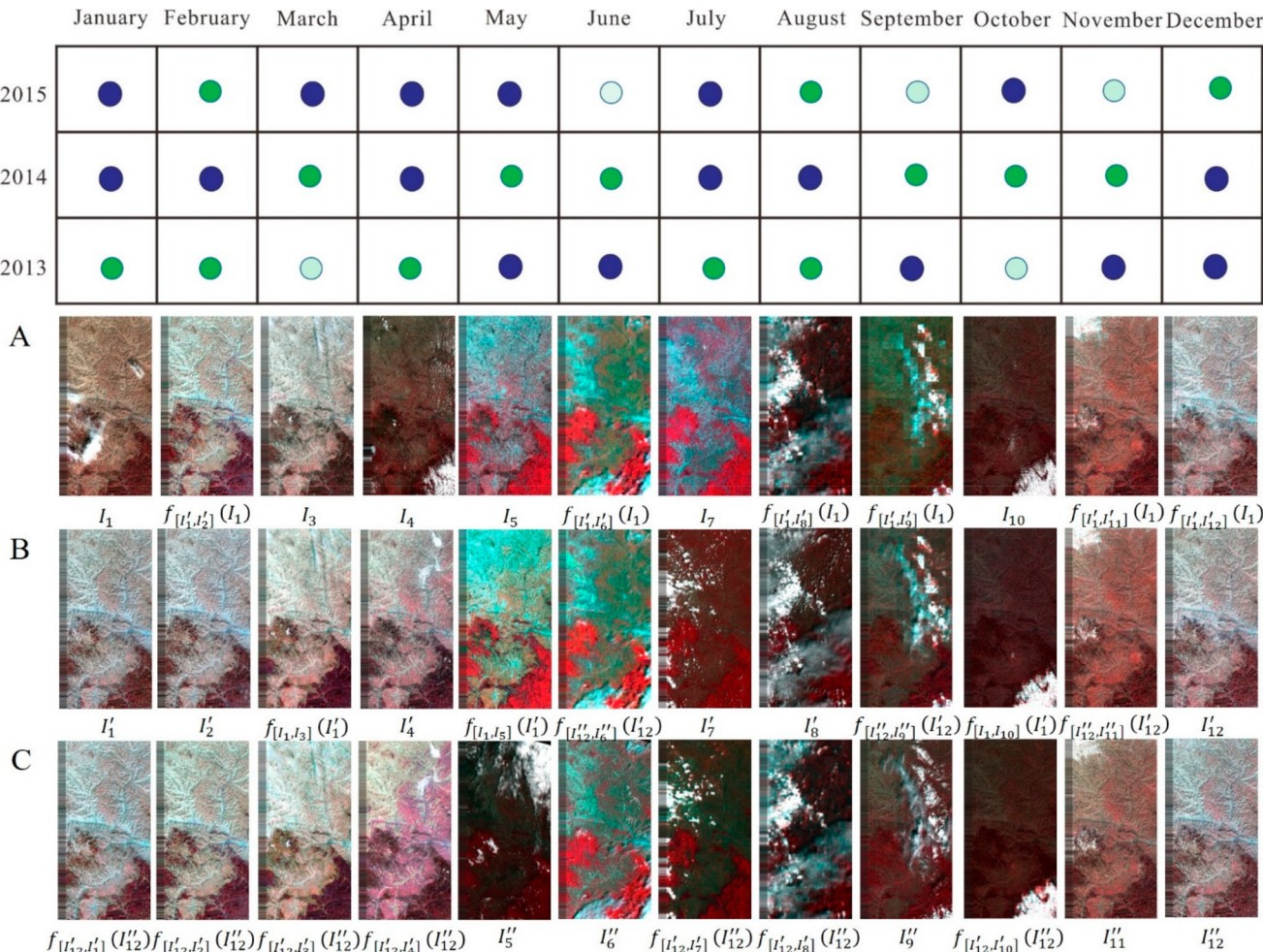

**Figure 11.** Generated results of Landsat-8 images using CyGAN network from 2013 to 2015 according to construction strategy in Table 4: existing images and generated results in (**A**) 2015 sequence, (**B**) 2014 sequence, and (**C**) 2013 sequence.

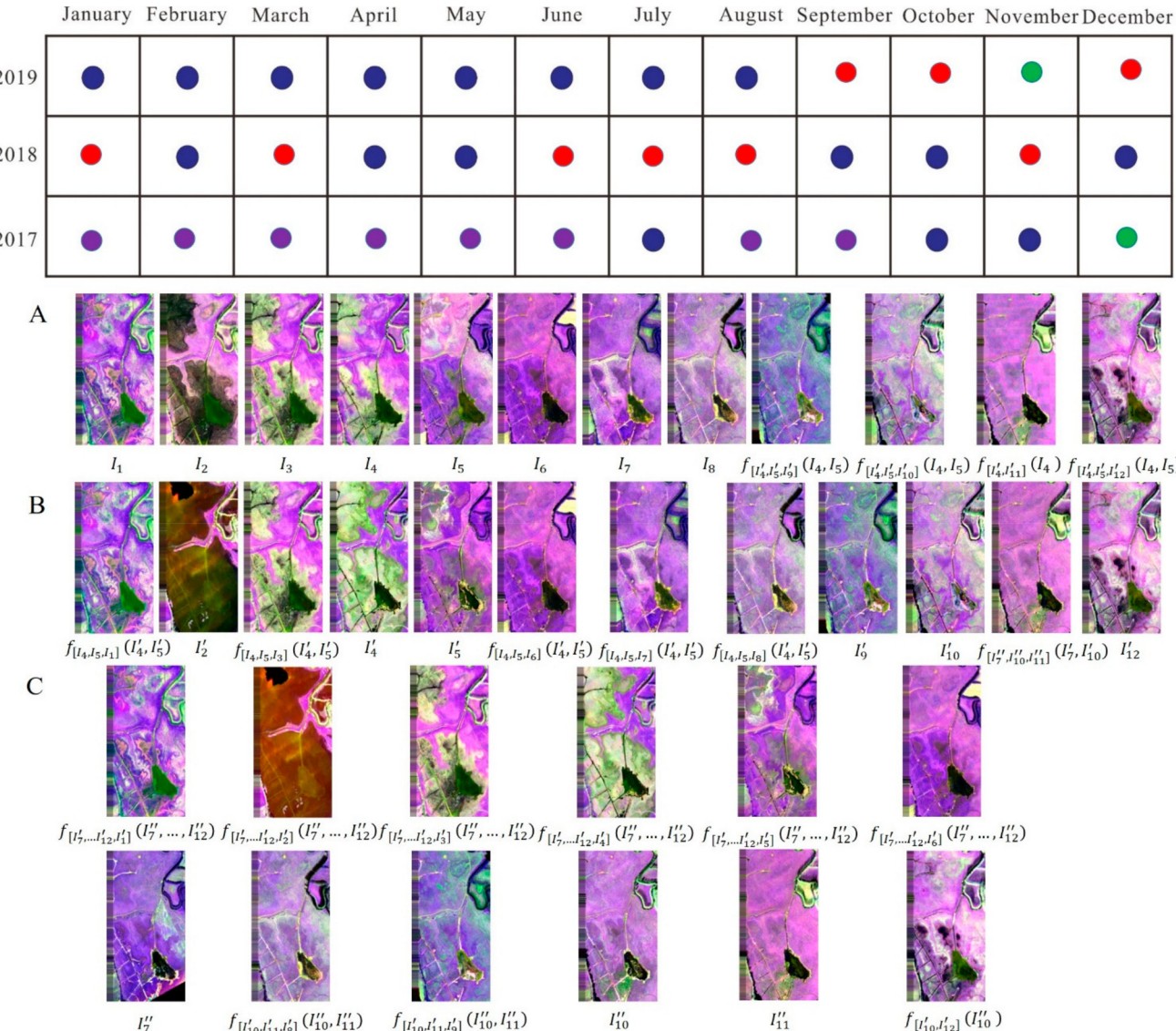

**Figure 12.** Generated results of UAV images using AdaFG, CLSTM, and CyGAN network from 2017 to 2019 according to construction strategy in Table 5: existing images and generated results in (**A**) 2019 sequence, (**B**) 2018 sequence, and (**C**) 2017 sequence.

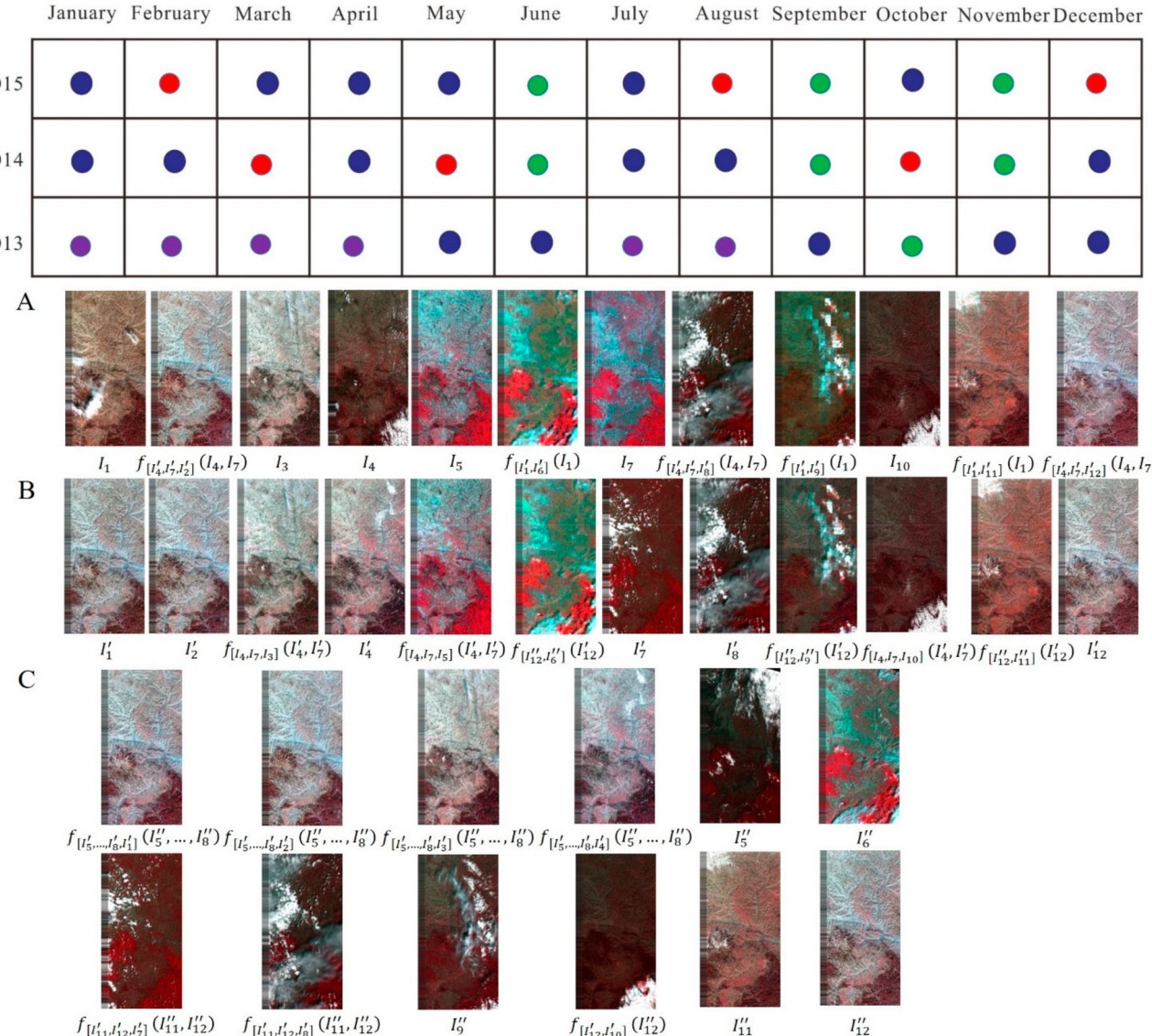

**Figure 13.** Generated results of Landsat-8 images using AdaFG, CLSTM, and CyGAN network from 2013 to 2015 according to construction strategy in Table 6: existing images and generated results in (**A**) 2015 sequence, (**B**) 2014 sequence, and (**C**) 2013 sequence.

### 3.3.4. Comparisons Between Single and Multiple Networks Datasets

In the second and third aspects, we conducted quantitative indicator evaluation between single network datasets and multiple network datasets. Table 8 shows the quantitative evaluation indicator between the generated results using a single network (AdaFG, CyGAN) and multiple networks. The table shows that the RMSE value produced by using a single network (AdaFG, CyGAN) was higher than that using multiple networks in both datasets.

**Table 8.** Quantitative evaluation indicator between the generated results using a single network (AdaFG, CyGAN) and multiple networks.

| Datasets | Single Network | Multiple Network | Reference Images | RMSE (Pixel) |
|---|---|---|---|---|
| UAV | $f_{[I'_4, I'_5, I''_{11}]}(I_4, I_5)$ | $f_{[I'_4, I''_{11}]}(I_4)$ | $I''_{11}$ | 4.953 4.702 |
| | $f_{[I'_{10}, I''_{11}, I'_{12}]}(I''_{10}, I''_{11})$ | $f_{[I'_{10}, I'_{12}]}(I''_{10})$ | $I'_{12}$ | 3.847 3.787 |
| | $f_{[I'_{10}, I''_{11}, I'_9]}(I''_{10}, I''_{11})$ | $f_{[I'_{10}, I''_{11}, I'_9]}(I''_{10}, I''_{11})$ | $I'_9$ | 2.723 2.638 |
| | $f_{[I'_{10}, I''_{11}, I'_8]}(I''_{10}, I''_{11})$ | $f_{[I'_{10}, I''_{11}, I'_8]}(I''_{10}, I''_{11})$ | $I_8$ | 2.879 1.933 |
| | $f_{[I'_{10}, I''_{11}, I'_6]}(I''_{10}, I''_{11})$ | $f_{[I'_7, \ldots, I'_{12}, I'_6]}(I''_7, \ldots, I''_{12})$ | $I_6$ | 3.483 1.228 |
| | $f_{[I'_{10}, I''_{11}, I'_5]}(I''_{10}, I''_{11})$ | $f_{[I'_7, \ldots, I'_{12}, I'_5]}(I''_7, \ldots, I''_{12})$ | $I'_5$ | 2.967 1.304 |
| | $f_{[I'_{10}, I''_{11}, I'_4]}(I''_{10}, I''_{11})$ | $f_{[I'_7, \ldots, I'_{12}, I'_4]}(I''_7, \ldots, I''_{12})$ | $I'_4$ | 3.100 1.664 |
| | $f_{[I'_{10}, I''_{11}, I'_3]}(I''_{10}, I''_{11})$ | $f_{[I'_7, \ldots, I'_{12}, I'_3]}(I''_7, \ldots, I''_{12})$ | $I_3$ | 3.864 2.704 |
| | $f_{[I'_{10}, I''_{11}, I'_2]}(I''_{10}, I''_{11})$ | $f_{[I'_7, \ldots, I'_{12}, I'_2]}(I''_7, \ldots, I''_{12})$ | $I'_2$ | 3.504 1.580 |
| | $f_{[I'_{10}, I''_{11}, I'_1]}(I''_{10}, I''_{11})$ | $f_{[I'_7, \ldots, I'_{12}, I'_1]}(I''_7, \ldots, I''_{12})$ | $I_1$ | 3.188 2.377 |
| Landsat-8 | $f_{[I'_1, I'_2]}(I_1)$ | $f_{[I'_4, I'_7, I'_2]}(I_4, I_7)$ | $I'_2$ | 2.695 1.144 |
| | $f_{[I'_1, I'_8]}(I_1)$ | $f_{[I'_4, I'_7, I'_8]}(I_4, I_7)$ | $I'_8$ | 5.017 1.170 |
| | $f_{[I'_1, I'_{12}]}(I_1)$ | $f_{[I'_4, I'_7, I'_{12}]}(I_4, I_7)$ | $I'_{12}$ | 2.777 0.929 |
| | $f_{[I_1, I_3]}(I'_1)$ | $f_{[I_4, I_7, I_3]}(I'_4, I'_7)$ | $I_3$ | 2.632 1.286 |
| | $f_{[I_1, I_5]}(I'_1)$ | $f_{[I_4, I_7, I_5]}(I'_4, I'_7)$ | $I_5$ | 3.375 1.411 |
| | $f_{[I_1, I_{10}]}(I'_1)$ | $f_{[I_4, I_7, I_{10}]}(I'_4, I'_7)$ | $I_{10}$ | 3.990 1.968 |
| | $f_{[I'_{12}, I'_8]}(I''_{12})$ | $f_{[I'_{11}, I'_{12}, I'_8]}(I''_{11}, I''_{12})$ | $I'_8$ | 5.022 1.841 |
| | $f_{[I'_{12}, I'_7]}(I''_{12})$ | $f_{[I'_{11}, I'_{12}, I'_7]}(I''_{11}, I''_{12})$ | $I'_7$ | 4.399 1.729 |
| | $f_{[I'_{12}, I'_4]}(I''_{12})$ | $f_{[I'_5, \ldots, I'_8, I'_4]}(I''_5, \ldots, I''_8)$ | $I'_4$ | 2.771 1.313 |
| | $f_{[I'_{12}, I'_3]}(I''_{12})$ | $f_{[I'_5, \ldots, I'_8, I'_3]}(I''_5, \ldots, I''_8)$ | $I_3$ | 3.367 2.140 |
| | $f_{[I'_{12}, I'_2]}(I''_{12})$ | $f_{[I'_5, \ldots, I'_8, I'_2]}(I''_5, \ldots, I''_8)$ | $I'_2$ | 2.606 2.122 |
| | $f_{[I'_{12}, I'_1]}(I''_{12})$ | $f_{[I'_5, \ldots, I'_8, I'_1]}(I''_5, \ldots, I''_8)$ | $I'_1$ | 2.686 2.629 |

## 4. Discussions

### 4.1. Separable Convolution Kernel Sizes

We observed that the separable convolution kernel size of the AdaFG network had an impact on the generated result, and conducted a visual comparison between the generated result and reference image using different separable convolution kernel sizes. We selected separable convolution kernels with sizes of 11, 13, and 15 to train the AdaFG network. Table 9 shows the quantitative evaluation indicator and Figure 14 the visual effect and pixel error between the generated result and reference image using different separable convolution kernel sizes.

**Table 9.** Quantitative evaluation between the generated result and reference image using different separable convolution kernel sizes.

| Datasets | Kernel Sizes | Generated Results | Reference Images | RMSE (Pixel) |
|---|---|---|---|---|
| UAV | 11 | $f_{[I_4,I_5,I_6]}\left(I'_4, I'_5\right)$ | $I_6$ | 1.128 |
| | | $f_{[I_4,I_5,I_7]}\left(I'_4, I'_5\right)$ | $I_7$ | 0.952 |
| | | $f_{[I_4,I_5,I_8]}\left(I'_4, I'_5\right)$ | $I_8$ | 0.954 |
| | 13 | $f_{[I_4,I_5,I_6]}\left(I'_4, I'_5\right)$ | $I_6$ | 1.437 |
| | | $f_{[I_4,I_5,I_7]}\left(I'_4, I'_5\right)$ | $I_7$ | 1.273 |
| | | $f_{[I_4,I_5,I_8]}\left(I'_4, I'_5\right)$ | $I_8$ | 1.416 |
| | 15 | $f_{[I_4,I_5,I_6]}\left(I'_4, I'_5\right)$ | $I_6$ | 1.444 |
| | | $f_{[I_4,I_5,I_7]}\left(I'_4, I'_5\right)$ | $I_7$ | 1.560 |
| | | $f_{[I_4,I_5,I_8]}\left(I'_4, I'_5\right)$ | $I_8$ | 1.418 |
| Landsat-8 | 11 | $f_{[I'_4,I'_7,I'_2]}\left(I_4, I_7\right)$ | $I'_2$ | 1.144 |
| | | $f_{[I'_4,I'_7,I'_8]}\left(I_4, I_7\right)$ | $I'_8$ | 1.170 |
| | | $f_{[I'_4,I'_7,I'_{12}]}\left(I_4, I_7\right)$ | $I'_{12}$ | 0.929 |
| | 13 | $f_{[I'_4,I'_7,I'_2]}\left(I_4, I_7\right)$ | $I'_2$ | 1.689 |
| | | $f_{[I'_4,I'_7,I'_8]}\left(I_4, I_7\right)$ | $I'_8$ | 2.533 |
| | | $f_{[I'_4,I'_7,I'_{12}]}\left(I_4, I_7\right)$ | $I'_{12}$ | 1.402 |
| | 15 | $f_{[I'_4,I'_7,I'_2]}\left(I_4, I_7\right)$ | $I'_2$ | 1.938 |
| | | $f_{[I'_4,I'_7,I'_8]}\left(I_4, I_7\right)$ | $I'_8$ | 2.875 |
| | | $f_{[I'_4,I'_7,I'_{12}]}\left(I_4, I_7\right)$ | $I'_{12}$ | 1.633 |

### 4.2. Stacked CLSTM Network Layers

We observed that the number of stacked CLSTM network layers had an impact on the generated result, and conducted a visual comparison between the generated result and reference image using different stacked CLSTM network layers. We selected $1 \times 1$, $2 \times 2$, and $3 \times 3$ stacked CLSTM network layers to train the CLSTM network. Table 10 shows the quantitative evaluation indicator and Figure 15 the visual effect and pixel error between the generated result and reference image using different stacked CLSTM network layers.

**Table 10.** Quantitative evaluation between the generated result and reference image using different stacked CLSTM network layers.

| Datasets | Stacked Numbers | Generated Results | Reference Images | RMSE (pixel) |
|---|---|---|---|---|
| UAV | $1 \times 1$ | $f_{[I_3,I_4,I_5,I_6]}\left(I'_3, I'_4, I'_5\right)$ | $I_6$ | 2.681 |
| | | $f_{[I_3,I_4,I_5,I_7]}\left(I'_3, I'_4, I'_5\right)$ | $I_7$ | 2.096 |
| | | $f_{[I_3,I_4,I_5,I_8]}\left(I'_3, I'_4, I'_5\right)$ | $I_8$ | 2.186 |
| | $2 \times 2$ | $f_{[I_3,I_4,I_5,I_6]}\left(I'_3, I'_4, I'_5\right)$ | $I_6$ | 1.976 |
| | | $f_{[I_3,I_4,I_5,I_7]}\left(I'_3, I'_4, I'_5\right)$ | $I_7$ | 1.556 |
| | | $f_{[I_3,I_4,I_5,I_8]}\left(I'_3, I'_4, I'_5\right)$ | $I_8$ | 1.617 |
| | $3 \times 3$ | $f_{[I_3,I_4,I_5,I_6]}\left(I'_3, I'_4, I'_5\right)$ | $I_6$ | 1.758 |
| | | $f_{[I_3,I_4,I_5,I_7]}\left(I'_3, I'_4, I'_5\right)$ | $I_7$ | 1.300 |
| | | $f_{[I_3,I_4,I_5,I_8]}\left(I'_3, I'_4, I'_5\right)$ | $I_8$ | 1.421 |
| Landsat-8 | $1 \times 1$ | $f_{[I'_4,I'_7,I'_2]}\left(I_4, I_7\right)$ | $I'_2$ | 2.265 |
| | | $f_{[I'_4,I'_7,I'_8]}\left(I_4, I_7\right)$ | $I'_8$ | 3.386 |
| | | $f_{[I'_4,I'_7,I'_{12}]}\left(I_4, I_7\right)$ | $I'_{12}$ | 1.935 |
| | $2 \times 2$ | $f_{[I'_4,I'_7,I'_2]}\left(I_4, I_7\right)$ | $I'_2$ | 1.780 |
| | | $f_{[I'_4,I'_7,I'_8]}\left(I_4, I_7\right)$ | $I'_8$ | 2.697 |
| | | $f_{[I'_4,I'_7,I'_{12}]}\left(I_4, I_7\right)$ | $I'_{12}$ | 1.577 |
| | $3 \times 3$ | $f_{[I'_4,I'_7,I'_2]}\left(I_4, I_7\right)$ | $I'_2$ | 1.537 |
| | | $f_{[I'_4,I'_7,I'_8]}\left(I_4, I_7\right)$ | $I'_8$ | 1.681 |
| | | $f_{[I'_4,I'_7,I'_{12}]}\left(I_4, I_7\right)$ | $I'_{12}$ | 1.484 |

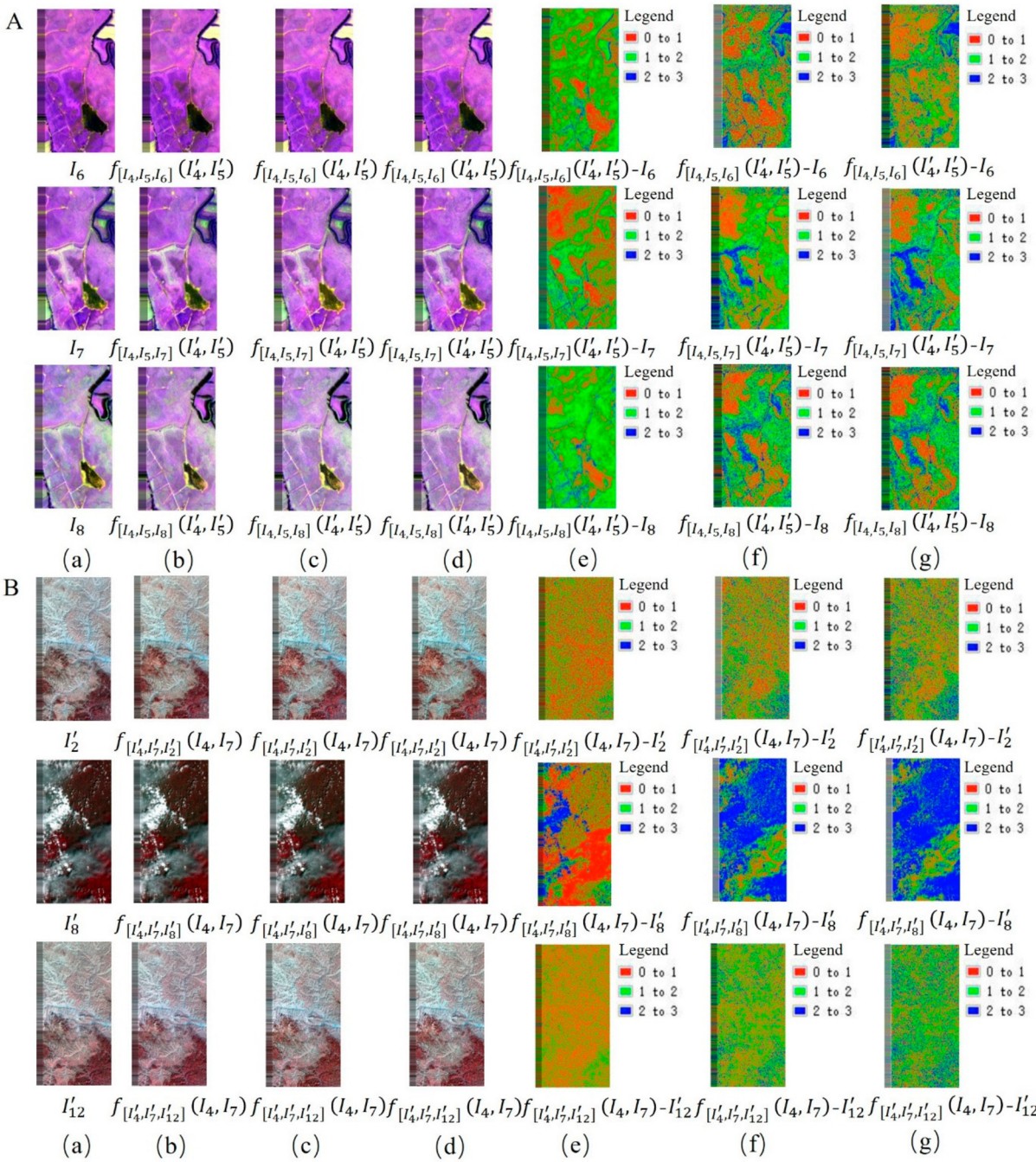

**Figure 14.** (**a**) Initial image, (**b**–**d**) Visual effect and (**e**–**g**) pixel error between the generated results and reference image using separable convolution kernel sizes 11, 13, and 15: (**A**) UAV and (**B**) Landsat-8 datasets.

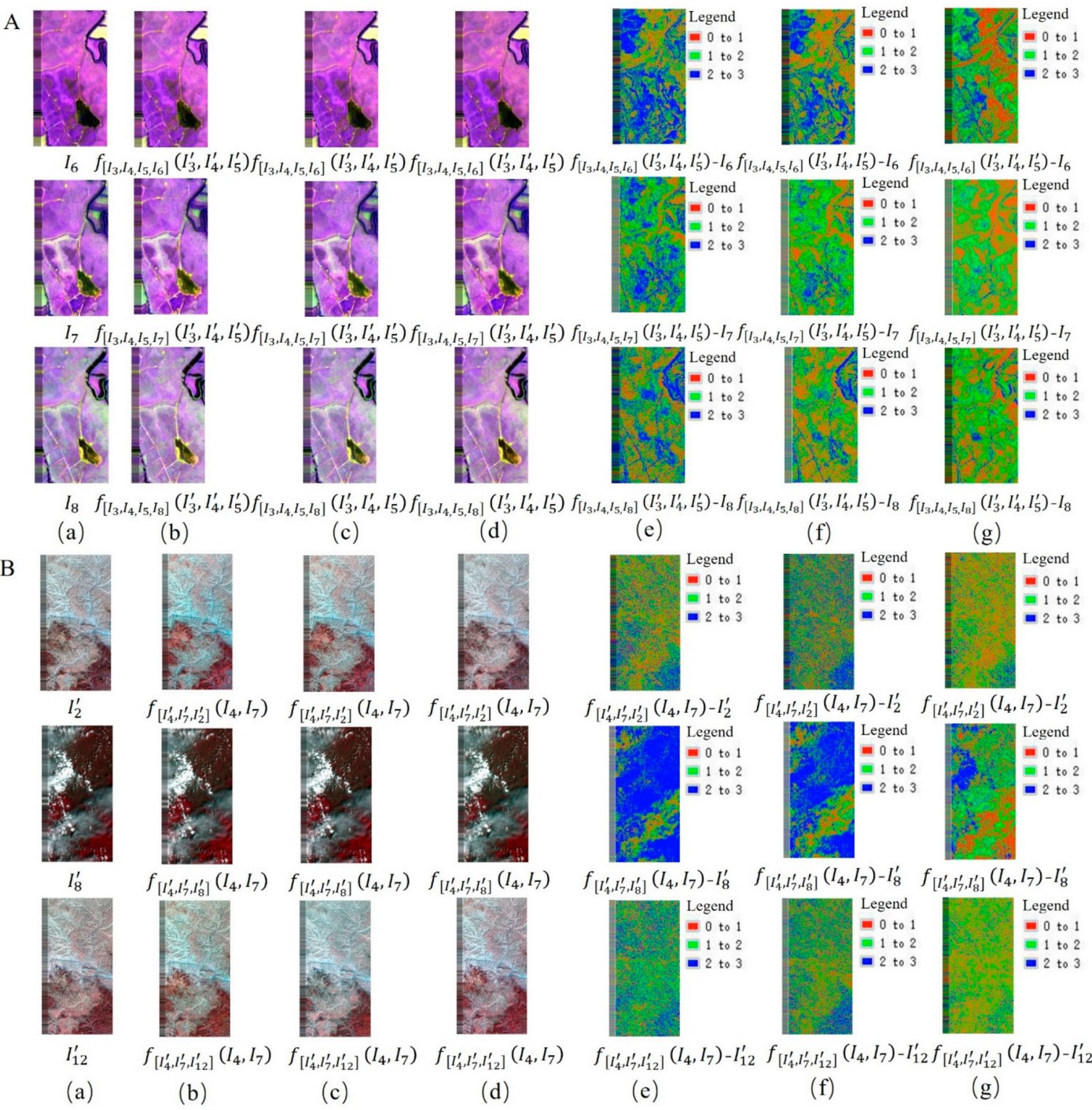

**Figure 15.** (**a**) Initial image, (**b**–**d**) Visual effect and (**e**–**g**) pixel error between the generated result and reference image using stacks of $1 \times 1$, $2 \times 2$, and $3 \times 3$ CLSTM network layers: (**A**) UAV and (**B**) Landsat-8 datasets.

### 4.3. Proportional Hyper-Parameters

We observed that the proportional hyper-parameters of the CyGAN network had an impact on the generated result, and conducted a visual comparison between the generated result and reference image using different proportional hyper-parameters. We selected $\mu_1 = 250, \mu_2 = 1/16, \mu_1 = 350, \mu_2 = 1/32$ and $\mu_1 = 450, \mu_2 = 1/64$ to train the CyGAN network. Table 11 shows the quantitative evaluation indicator and Figure 16 visual effect and pixel error between the generated result and reference image using different proportional hyper-parameters.

**Table 11.** Quantitative evaluation between the generated result and reference image using different proportional hyper-parameters.

| Datasets | $\mu_1, \mu_2$ | Generated Results | Reference Images | RMSE (Pixel) |
|---|---|---|---|---|
| UAV | 250,1/16 | $f_{[I_4, I_6]}(I'_4)$ | $I_6$ | 3.643 |
| | | $f_{[I_4, I_7]}(I'_4)$ | $I_7$ | 3.271 |
| | | $f_{[I_4, I_8]}(I'_4)$ | $I_8$ | 3.836 |
| | 350,1/32 | $f_{[I_4, I_6]}(I'_4)$ | $I_6$ | 3.585 |
| | | $f_{[I_4, I_7]}(I'_4)$ | $I_7$ | 3.077 |
| | | $f_{[I_4, I_8]}(I'_4)$ | $I_8$ | 3.690 |
| | 450,1/64 | $f_{[I_4, I_6]}(I'_4)$ | $I_6$ | 3.814 |
| | | $f_{[I_4, I_7]}(I'_4)$ | $I_7$ | 3.263 |
| | | $f_{[I_4, I_8]}(I'_4)$ | $I_8$ | 3.698 |
| Landsat-8 | 250,1/16 | $f_{[I'_1, I'_2]}(I_1)$ | $I'_2$ | 2.824 |
| | | $f_{[I'_1, I'_8]}(I_1)$ | $I'_8$ | 5.708 |
| | | $f_{[I'_1, I'_{12}]}(I_1)$ | $I'_{12}$ | 2.875 |
| | 350,1/32 | $f_{[I'_1, I'_2]}(I_1)$ | $I'_2$ | 2.694 |
| | | $f_{[I'_1, I'_8]}(I_1)$ | $I'_8$ | 5.017 |
| | | $f_{[I'_1, I'_{12}]}(I_1)$ | $I'_{12}$ | 2.629 |
| | 450,1/64 | $f_{[I'_1, I'_2]}(I_1)$ | $I'_2$ | 2.695 |
| | | $f_{[I'_1, I'_8]}(I_1)$ | $I'_8$ | 5.553 |
| | | $f_{[I'_1, I'_{12}]}(I_1)$ | $I'_{12}$ | 2.777 |

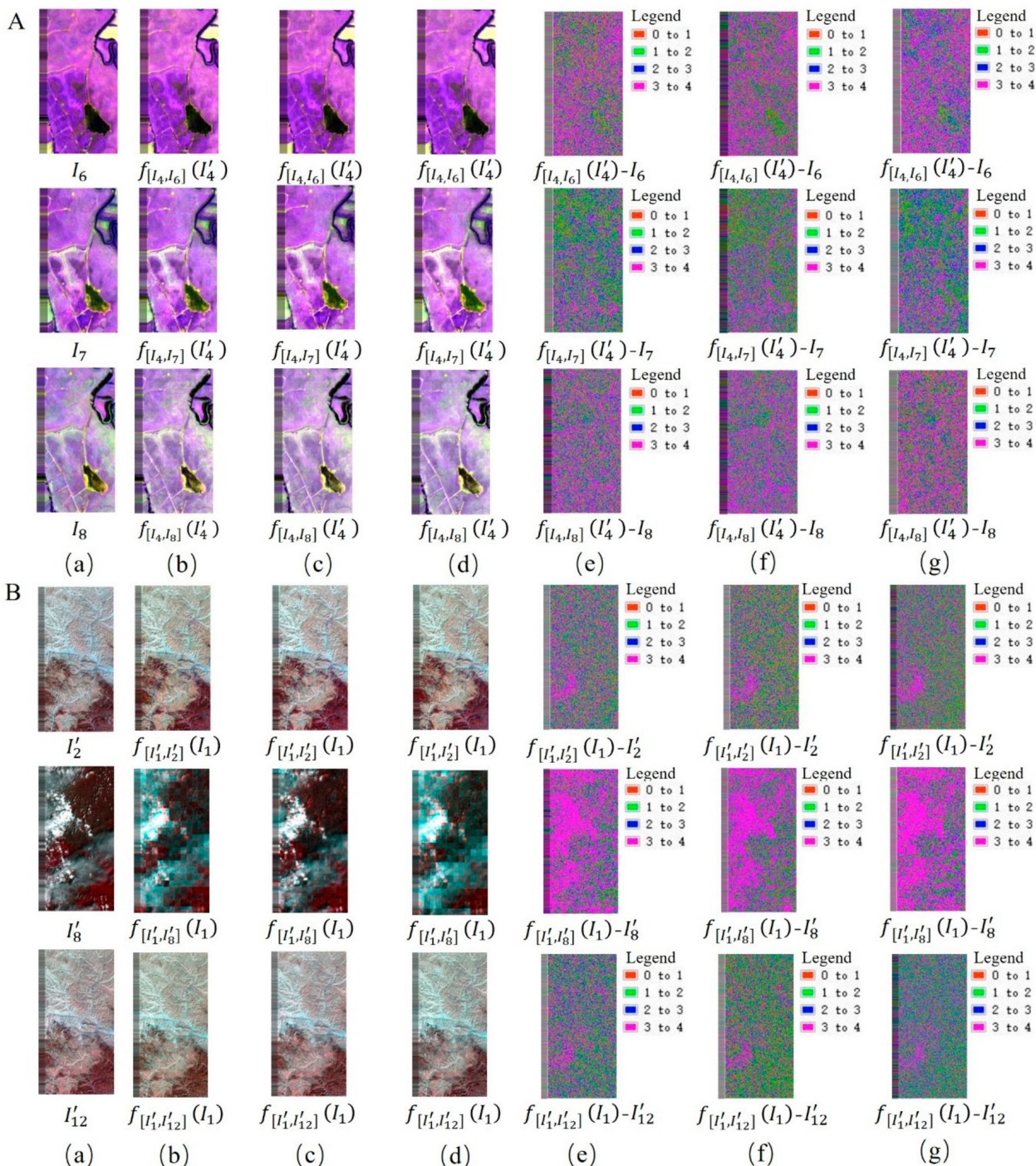

**Figure 16.** (**a**) Initial image, (**b**–**d**) Visual effect and (**e**–**g**) pixel error between the generated result and reference image using different proportional hyper-parameters: (**A**) UAV and (**B**) Landsat-8 datasets.

## 5. Conclusions

The paper provides a remote-sensing sequence image construction scheme that can transform spectral mapping estimation and pixel synthesis into an easier process of using different deep convolution networks to construct remote-sensing sequence datasets. The major conclusions of this paper can be summarized as follows:

(1) The proposed AdaFG model provides a new method of generating remote-sensing images, especially for high-spatial-resolution images. CLSTM and CyGAN models can make up for the shortcoming of the AdaFG network model, which only performs single-

scene generation and is affected by the missing degree of sequence images. The models can better capture and simulate complex and diverse nonlinear spectral transformation between different temporal images, and get better-generated images based on the models.

(2) Using separable convolution kernel of size 11 in AdaFG network, stacks of $3 \times 3$ CLSTM network layer, and $\mu_1 = 350$, $\mu_2 = 1/32$ in CyGAN network lead to better-generated results. Experiments show that the deep convolution network models can be used to get generated images to fill in missing areas of sequence images and produce remote-sensing sequence datasets.

The proposed AdaFG model is an image transformation method based on triple samples according to its structure, and its structure uses an adaptive separable convolution kernel. The spectral transformation ability of this model is higher than CLSTM and CyGAN models. However, this method only performs single-scene generation and its results are affected by the missing degree of sequence images. The proposed CLSTM model is an image transformation method based on multi-tuple samples according to its structure, and its structure is single-step stacked prediction. This model can perform predictions based on the state information of the time-series data. However, its results are affected by the length and missing degree of sequence images. The proposed CyGAN model is an image transformation method based on two-tuple samples according to its structure, and its structure uses cycle-consistent strategy and variable proportional hyper-parameters. This model can perform bidirectional transformation between different pictures, and its generated results are not affected by the degree of missing sequence images. However, the spectral transformation ability of this method is weak and its results are affected by the noise degree of sequence images.

When constructing remote-sensing time-series image datasets, it is necessary to make full use of the advantages of different deep convolution networks. When the missing degree of the sequence is small, using AdaFG and CLSTM model constructs the construction of remote-sensing time-series image datasets. When the missing degree of the sequence is larger, using the CyGAN model constructs the construction of remote-sensing time-series image datasets. Regardless of the quality and missing degree of sequence images, it is necessary to use the AdaFG model for the construction of remote-sensing time-series image datasets. In future work, we will make full use of the advantages of different deep convolution networks to construct a larger area remote-sensing sequence datasets.

**Author Contributions:** Conceptualization, P.T.; Methodology, X.J.; Writing—original draft, X.J.; Writing—review & editing, P.T.; funding acquisition, Z.Z. All authors have read and agreed to the published version of the manuscript.

**Funding:** This work was supported by the Strategic Priority Research Program of the Chinese Academy of Sciences (grant no. XDA19080301) and the National Natural Science Foundation of China (grant no. 41701399).

**Data Availability Statement:** Data sharing does not apply to this article.

**Acknowledgments:** The authors would like to thank Université de Rennes 2 for providing the UAV datasets. We would like to thank the editors and the reviewers for their detailed comments and suggestions, which greatly helped us to improve the clarity and presentation of our manuscript.

**Conflicts of Interest:** The authors declare no conflict of interest.

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
