# Peer review of "Sequence Image Datasets Construction via Deep Convolution Networks"

_remotesensing, doi:10.3390/rs13091853_

Round 1

Reviewer 1 Report

This paper applies deep convolutional neural networks to learn the complex mapping between sequence images for the construction of sequence image datasets. 

The method is inspired by the idea of Niklaus, Pan, Zhu et al., and exploits the advantages of different deep convolutional neural networks to make up for the missing remote-sensing data.

The proposed AdaFG network model provides a new method of generating remote-sensing images for high-spatial-resolution images.
CyGAN and CLSTM models can make up for the shortcoming of the AdaFG network model, which only performs single-scene generation.
The models are able to capture and simulate complex and diverse nonlinear spectral transformation between different temporal images and achieve better-generated images.

Experiments performed with unmanned aerial vehicle (UAV) and Landsat-8 datasets show that the deep neural networks are effective to produce high-quality time-series image datasets, and the data-driven deep convolution networks can better simulate complex and diverse nonlinear data information.

The paper is well written, the approach is interesting and the results are promising. 

My only concern with the paper is that it goes straight to the method and the experiments, without a comprehensive analysis of the literature.
The introduction, in fact, consists of only one page. Deep learning methods that involve data augmentation is very popular nowadays, and I believe that the introduction overlooks many of the available approaches (see 10.3390/rs13040547, 10.3390/rs13040659, 10.3390/rs12244142). I recommend the authors extend this section, including the proposed papers and possibly extending with additional ones. This adaptation will provide a more comprehensive overview to the reader and the opportunity to discuss how the proposed method can position itself in the landscape of approaches in the literature.

Author Response

Thank you very much for your constructive suggestions. Please see the attachment.

Reviewer 2 Report

Congratulations for the presented manuscript. It addresses an interesting research topic with relevant methodology. In particular, you compare different neural network architectures to evaluate the completeness of image sequence. From my point of view, just few comments I could add:

  • Indicate in the introduction why the three types of neural networks have been used (features that match for this application).
  • Indicate a reference for each neural network in Sections 2.2.1, 2.2.2, 2.2.3
  • Footers bellow the images in each Figure (e.g. f(...)) are really small, and it is difficult to read. Could you increase the font size?
  • You suggest the use of other neural network architectures because the used ones have certain limitations. Could you correlate those limitations with their architecture design (reason)? Taking in consideration these conclusions, could you suggest which new architectures do you expect to be better for this application?

Author Response

(The authors gave the same response as above.)

Reviewer 3 Report

The authors present an article in which they consider the deep convolution networks to learn the complex mapping between sequence images, showing that the deep convolution networks are effective to produce high-quality time-series image datasets, and the data-driven deep convolution networks can better simulate complex and diverse nonlinear data information. The article is well written and presented, showing sound results. 

I would like to ask the authors to discuss whether Generative Adversarial Networks are also applicable to produce this type of image datasets, and if so, I would like them to include some references related to them in the Introduction section, for instance [1] and [2].

Regarding the presentation of the article, I would like to ask the authors to make sure that there are not page breaks between a section title and its content, or a table header and the table, as it happens in Section 2, Table 1, Table 5, Section 3.2 and so on.

Besides, there are some minor typos so the paper should be carefully revised, such as page 9, lines 213 and 214 “The corresponding experimental data were described in the following paragraphs in detail.” “Were” should be changed by “are” or “will be” or same page, line 216 “We represents” should be changed by “We represent”.

Author Response

(The authors gave the same response as above.)

Round 2

Reviewer 1 Report

The authors extensively addressed the comments provided in the previous round of reviews. I believe that the paper has improved and it is now suitable for publication.